# Spotlight on Token Perception for Multimodal Reinforcement Learning

**Siyuan Huang**[1,2]**, Xiaoye Qu**[1†]**, Yafu Li**[3]**, Yun Luo**[1]**,**
**Zefeng He**[4]**, Daizong Liu**[5]**, Yu Cheng**[3†]
[1]Shanghai AI Laboratory, [2]Shanghai Jiao Tong University,
[3]The Chinese University of Hong Kong, [4]Nanjing University, [5]Wuhan University
`{huangsiyuan2, quxiaoye}@pjlab.org.cn, chengyu@cse.cuhk.edu.hk`

## Abstract

While Reinforcement Learning with Verifiable Rewards (RLVR) has advanced the reasoning capabilities of Large Vision-Language Models (LVLMs), most existing methods in multimodal reasoning neglect the critical role of visual perception within the RLVR optimization process. In this paper, we undertake a pioneering exploration of multimodal RLVR through the novel perspective of token perception, which measures the visual dependency of each generated token. With a granular analysis of Chain-of-Thought (CoT) processes, we uncover two key insights: first, token perception in a rollout trajectory is sparsely distributed, where only a small fraction of tokens have high visual dependency for visually-grounded reasoning; second, different trajectories exhibit significant divergence in their overall visual dependency. Based on these observations, we propose **V**isually-**P**erceptive **P**olicy **O**ptimization (**VPPO**), a novel policy gradient algorithm that explicitly leverages token perception to refine the learning signal. Specifically, VPPO achieves this through a dual mechanism: it reweights a trajectory's advantage by its overall visual dependency, and focuses policy updates exclusively on perceptually pivotal tokens. On a comprehensive suite of eight perception and reasoning benchmarks, VPPO demonstrates substantial gains over leading open-source RL-tuned models, with its effectiveness consistently validated across 7B and 32B model scales. Our findings not only establish a new token-level perceptual perspective for analyzing multimodal RLVR but also present a novel and effective optimization strategy to significantly enhance the multimodal reasoning capabilities of LVLMs. Our code is available at `https://github.com/huaixuheqing/VPPO-RL`.

## 1 Introduction

Reinforcement learning from verifiable rewards (RLVR) (Zhang et al., 2025), particularly with online algorithms like Group Relative Policy Optimization (GRPO), has dramatically advanced the reasoning capabilities of Large Language Models (LLMs) in text-centric domains, such as math and code (Shao et al., 2024; Guo et al., 2025; OpenAI, 2024; Team et al., 2025; Yang et al., 2025a; Anthropic, 2025). Recently, many works have attempted to translate this success to Large Vision-Language Models (LVLMs). These efforts primarily focus on three directions: data-centric enhancements (Li et al., 2025; Liang et al., 2025; Liu et al., 2025a; Yao et al., 2025; Chen et al., 2025a; Meng et al., 2025; Huang et al., 2025; Yang et al., 2025b), reward-centric engineering (Shen et al., 2025; Xia et al., 2025; Wang et al., 2025b; Xiao et al., 2025; Yu et al., 2025a; Wan et al., 2025), and other algorithmic adjustments (Wang et al., 2025a; Zhao et al., 2025).

However, prevailing RLVR frameworks for LVLMs largely neglect the critical role of visual perception in the optimization process. Effective reasoning is contingent upon accurate perception, which provides the essential grounding for logical deduction (Xiao et al., 2025). The geometry problem in Figure 1 exemplifies this dependency. Given a question: "In circle $\odot O$, $AC$ is parallel to $OB$, and $\angle BOC = 50°$. What is the measure of $\angle OAB$?" To correctly answer this question, a critical

---

[†]Corresponding authors.

insight should be derived from the visual diagram, namely segments $OA$ and $OB$ are radii of the circle $\odot O$, rendering $\triangle AOB$ isosceles. Therefore, without explicitly integrating perceptual ability into the core learning objectives, models cannot develop genuine multimodal reasoning capabilities (Yu et al., 2025a; Xiao et al., 2025).

In this paper, we analyze the perceptual mechanisms of multimodal RLVR through an innovative lens of token perception, investigating the impact of tokens with varying visual dependency on reasoning. With a granular analysis, we first point out that in the Chain-of-Thought (CoT) (Wei et al., 2022) processes of multimodal reasoning, the token perception distribution in a rollout trajectory exhibits a distinct pattern, where the majority of tokens are generated with low visual dependency, while a critical minority of tokens emerge with high dependency. After aggregating the token perception at the trajectory level, we further observe that different reasoning trajectories also exhibit significant divergence in their overall perceptual quality, as only a part of trajectories are genuinely perception-driven paths. Although those paths without significant visual perception may still fortuitously arrive at the correct answer, the resulting models will exhibit weak multimodal perception capabilities. These observations pinpoint a foundational flaw inherited from text-based RLVR, i.e, existing implementations directly train over all tokens with limited understanding of which tokens actually facilitate multimodal perception and reasoning. The indiscriminate broadcasting of a single, coarse reward to every trajectory and token hinders further performance gains by failing to prioritize critical perception-related trajectories and tokens.

Building upon the above discovery of token perception, we introduce **V**isually-**P**erceptive **P**olicy **O**ptimization (**VPPO**), a novel policy gradient algorithm to explicitly integrate the token perception into the policy update of multimodal RL, as illustrated in Figure 1. Specifically, our VPPO first quantifies the visual dependency of each token. Based on this visual dependency, we devise two strategies. First, to align the learning objective with perception-grounded trajectories, VPPO reweights each trajectory's advantage using its average dependency. In this way, the learning signal is steered toward robust, perception-grounded reasoning paths over spurious shortcuts. Second, to focus the learning signal on what truly mat-

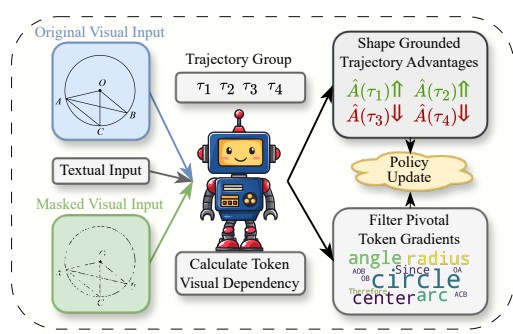

Figure 1: Our VPPO framework explicitly relies on token visual dependency to shape trajectory advantages and filter token gradients.

ters, VPPO constructs a sparse gradient mask to concentrate policy updates exclusively on critical visually-grounded reasoning tokens. This directly counters signal dilution, yielding a lower-variance gradient that leads to faster convergence and a stronger final policy. Notably, our VPPO can be seamlessly plugged into mainstream RLVR algorithms such as GRPO and DAPO.

To validate the effectiveness of our proposed VPPO, we conduct extensive experiments across a suite of eight challenging multimodal reasoning benchmarks, covering mathematical, geometric, logical, and multi-discipline reasoning. Based on Qwen2.5-VL series models, our 7B variant achieves a remarkable 19.2% average accuracy improvement over baseline, also surpassing previous open-source leading methods. This robust performance seamlessly scales to the 32B model, which also brings a 7.6% average accuracy improvement. Crucially, these performance gains are achieved alongside superior training stability and faster convergence, underscoring its efficiency and robustness.

To sum up, our main contributions are threefold:

- In this paper, we make the first attempt to analyze the perceptual mechanisms of multimodal RLVR through an innovative lens of token perception. We discover that only a critical minority of tokens emerge with high visual dependency, while only a part of the trajectories are genuinely perception-driven paths.
- We introduce VPPO, a novel policy gradient algorithm that explicitly focuses on token perception, leveraging visual dependency to align trajectory-level objectives and focus token-level gradient updates. In this way, the model spotlights perception while reasoning.
- Our extensive experiments on eight perception and reasoning benchmarks demonstrate our VPPO's superior performance. We further show its robust scalability across 7B and 32B model scales. We also perform in-depth ablation studies to validate the critical designs in our VPPO.

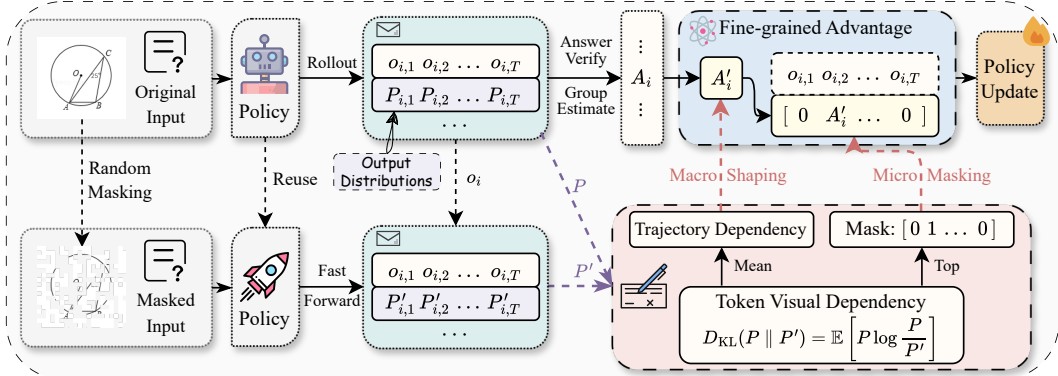

Figure 2: Overview of our VPPO framework. Given the original and masked image inputs, we first obtain the corresponding output distributions. Then, we compute a token-level visual dependency score for each trajectory. Subsequently, these token-level scores are used to generate two hierarchical control signals: at the macro-level, they are averaged into a trajectory-level dependency to shape the advantage, while at the micro-level, the top-$k\%$ tokens are identified to create a sparse binary token gradient mask. In this way, the uniform advantage is transformed into a fine-grained, targeted learning signal for the final policy update.

## 2 RELATED WORK

**Multimodal Reasoning.** While LLMs have achieved powerful reasoning in text-only domains (Guo et al., 2025), their visual counterparts, Large Vision-Language Models (LVLMs) (Bai et al., 2025a; Hurst et al., 2024; Team et al., 2024), still exhibit a significant performance gap when tasked with this complex integration (Wang et al., 2024b; Dong et al., 2025; Fang et al., 2024b;a; 2026; Liu et al., 2024b;a). Bridging this gap requires adapting the reasoning successes from text-only models to the unique demands of the multimodal space, where foundational algorithms like PPO (Schulman et al., 2017) and GRPO (Shao et al., 2024) are being actively explored.

**Dominant Strategies in Multimodal RL.** Most strategies focus on enhancing components external to the core learning algorithm. These approaches are largely either data-centric, focusing on the curation of visually-grounded datasets (Bai et al., 2025b; Li et al., 2025; Liang et al., 2025), distillation of Chain-of-Thought data (Chen et al., 2025b; Huang et al., 2025; Meng et al., 2025), and design of training curricula (Chen et al., 2025c; Wei et al., 2025); or reward-centric, seeking to engineer more informative, perception-aware signals (Wang et al., 2025e; Ma et al., 2025; Fan et al., 2025; Liu et al., 2025b; Yang et al., 2025b; Xia et al., 2025; Chen et al., 2025d; Wan et al., 2025). Other tactics include modifying rollouts or integrating external vision tools (Liu et al., 2025a; Wang et al., 2025a; Zheng et al., 2025b). While modality-agnostic algorithmic advances like Dynamic Sampling Policy Optimization (DAPO) (Yu et al., 2025b) introduce effective techniques like dynamic sampling and clip-higher, they still broadcast a uniform learning signal to all tokens. Our VPPO counters this core limitation by intervening internally, using visual dependency to reweight trajectory advantages and focus gradient updates on pivotal moments of visually-grounded reasoning.

**Pivotal Tokens in Reasoning.** Prior works in RL for large language models identify the pivotal tokens via high-entropy "forking points" (Wang et al., 2025c), low-confidence error points targeted for exploration (Vassoyan et al., 2025), or contrastive estimation between models trained on correct vs. incorrect data (Lin et al., 2024). However, for the multimodal domain, a pivotal token is not merely a logical fork but a critical moment of visually-grounded reasoning. In this paper, we introduce VPPO, the first multimodal RL algorithm designed to formally identify the perceptually pivotal tokens via dependency and then leverage them for targeted optimization.

## 3 METHOD

In this paper, as shown in Figure 2, we introduce **V**isually-**P**erceptive **P**olicy **O**ptimization (**VPPO**) that explicitly focuses on token perception by hierarchically shaping trajectory-level advantages and filtering token-level gradients. This targeted signal modulation fosters more stable, efficient, and interpretable learning.

## 3.1 PRELIMINARY: GROUP RELATIVE POLICY OPTIMIZATION (GRPO)

Given a multimodal prompt $(I, q)$ consisting of a visual input $I$ and a textual query $q$, the old policy $\pi_{\theta_{old}}$ generates a group of $G$ responses, $\{o_i\}_{i=1}^G$. In the RLVR framework, a binary reward $R_i \in \{0, 1\}$ is assigned to each complete response based solely on whether its final extracted answer matches the ground truth. While GRPO mitigates reward sparsity through a group-based advantage estimation, it remains fundamentally reliant on this coarse, outcome-based signal.

The advantage $\hat{A}_i$ for a response $o_i$ is its normalized reward:

$$\hat{A}_i = \frac{R_i - \text{mean}(\{R_k\}_{k=1}^G)}{\text{std}(\{R_k\}_{k=1}^G)} \tag{1}$$

The policy $\pi_\theta$ is then updated to maximize a clipped surrogate objective, where this uniform advantage $\hat{A}_i$ is broadcast to every timestep $t$:

$$\mathcal{L}^{\text{GRPO}}(\theta) = \mathbb{E}\left[ \frac{1}{G} \sum_{i=1}^G \frac{1}{|o_i|} \sum_{t=1}^{|o_i|} \min\left( r_{i,t}(\theta)\hat{A}_i, \text{clip}(r_{i,t}(\theta), 1 - \varepsilon, 1 + \varepsilon)\hat{A}_i \right) \right] \tag{2}$$

where $r_{i,t}(\theta) = \frac{\pi_\theta(o_{i,t}|I,q,o_{i,<t})}{\pi_{\theta_{old}}(o_{i,t}|I,q,o_{i,<t})}$ is the probability ratio.

While scalable, this outcome-based verification introduces a two-tiered limitation as follows:

1. **Trajectory-Level Ambiguity:** It treats all correct solutions equally, failing to distinguish a reasoning path that is strongly grounded in visual evidence from one that arrives at the same answer through linguistic priors or hallucination.

2. **Token-Level Uniformity:** The single, coarse reward is then applied indiscriminately to every token in the sequence, failing to selectively reward the specific, pivotal moments of visually-grounded reasoning that led to the correct outcome.

## 3.2 VISUALLY-PERCEPTIVE POLICY OPTIMIZATION (VPPO)

To study the perception in multimodal reasoning, we first develop a metric to quantify visual dependency at each token and analyze the token perception in Section 3.2.1. Subsequently, based on the token perception, we further aggregate them into the trajectory-level dependency and uncover key insights into their non-uniform nature in Section 3.2.2. Based on these findings, we introduce VPPO in Section 3.2.3 for perception-centric multimodal reasoning.

### 3.2.1 QUANTIFYING TOKEN VISUAL DEPENDENCY

We define a token's visual dependency as the information gain provided by the visual context. This is quantified by computing the Kullback-Leibler (KL) divergence between the policy's predictive distribution conditioned on the true image versus a perturbed version, formally measuring the distributional shift attributable to visual input. The choice of KL divergence is validated in Appendix G, where it outperforms other metrics like Jensen-Shannon Divergence and simple probability shifts.

**Definition 3.1 (Token-level visual dependency).** *Let $I$ be the visual input and $I'$ be a non-informative, perturbed version. At a given state $s_t = (q, o_{<t})$, the visual dependency $\mathcal{S}$ at step $t$ is the KL divergence between the policy's output distributions conditioned on $I$ and $I'$:*

$$\mathcal{S}(s_t, I) := D_{KL}\left( \pi_\theta(\cdot|s_t, I) \| \pi_\theta(\cdot|s_t, I') \right). \tag{3}$$

*A high $\mathcal{S}$ value indicates that the image provides critical information for the token prediction at step $t$, marking it as a key moment of visually-grounded reasoning.*

With the above metric measuring the visual dependency for each token, we analyze the empirical distribution of token perception. To achieve this, we perform inference with the Qwen2.5-VL-7B model on the vision-dominant subset of the MathVerse (Zhang et al., 2024) benchmark. We then compute the token visual dependency for every token across all generated trajectories and demonstrate their frequency distribution in Figure 3. The y-axis is on a logarithmic scale to better visualize the distribution's long tail, and a Kernel Density Estimation (KDE) curve is overlaid for easier visualization of the trend. This analysis leads to our first key insight:

**Insight 1: Token Visual Dependency is Sparsely Distributed.** Within the trajectory, visual reasoning is driven by a sparse set of pivotal tokens. Figure 3 shows the sparse distribution of token-level visual dependency. Plotted on a logarithmic y-axis, the frequency drops exponentially as dependency increases. This highly skewed distribution confirms that only a small fraction of tokens are critical for visually-grounded reasoning. Further analysis confirms their semantic importance, as these high-dependency tokens predominantly consist of numbers, geometric concepts, and logical operators essential for the reasoning process. Broadcasting a uniform learning signal to all tokens thus dilutes the reward by rewarding many irrelevant, non-perceptual steps.

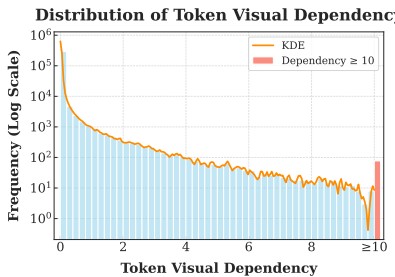

Figure 3: The skewed distribution of token-level visual dependency.

### 3.2.2 ANALYSIS OF REASONING TRAJECTORIES

After analyzing the token-level dependency, we aggregate this metric to the trajectory level by defining the trajectory dependency $\bar{S}(\tau)$ as the mean of the token-level dependency scores over a full trajectory $\tau$. This score represents the trajectory's overall reliance on visual evidence. To explore its distribution, we use the same experimental setup as before, plotting the frequency of these trajectory dependency scores in Figure 4. This reveals our second key insight:

**Insight 2: Trajectories Exhibit Heterogeneous Visual Grounding.** Not all correct reasoning paths are created equal. As shown in Figure 4, the distribution of trajectory-level visual dependency is heterogeneous. While loosely Gaussian, the distribution is right-skewed with a long tail, revealing that a distinct subset of high-dependency trajectories pulls the mean (0.09) to the right of the distribution's peak. Standard RL frameworks, by assigning a uniform reward, fail to distinguish the high perceptual informativeness of these trajectories, and thus cannot preferentially learn from genuine visually-grounded reasoning.

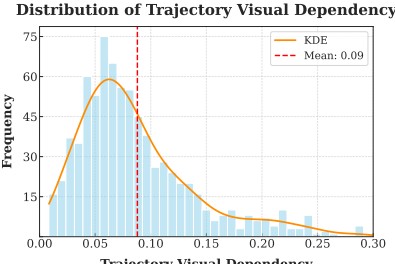

Figure 4: Distribution of trajectory dependency on perception.

### 3.2.3 VPPO POLICY GRADIENT ALGORITHM

Based on these insights, we introduce VPPO, a novel gradient algorithm that reshapes the learning signal at two levels of granularity to explicitly focus on token perception.

**Micro-level: Token-level Gradient Filtering (TGF).** Inspired by **Insight 1**, we focus on the learning signal exclusively on pivotal tokens. For each trajectory $\tau_i$, we identify the set of indices $\mathcal{K}_i$ corresponding to the top-$k\%$ of tokens with the highest visual dependency scores. This set defines a binary gradient mask $m_{i,t}$:

$$m_{i,t} = \mathbb{I}(t \in \mathcal{K}_i) = \begin{cases} 1 & \text{if token } t \text{ is a pivotal visual-reasoning token} \\ 0 & \text{otherwise} \end{cases} \tag{4}$$

This mask ensures that policy gradients are computed only for the pivotal tokens that bridge vision and language, effectively filtering out noise from generic tokens and combating signal dilution.

**Macro-level: Trajectory-level Advantage Shaping (TAS).** Inspired by **Insight 2**, we prioritize learning from superior, high-dependency trajectories. We compute a shaping factor $\alpha(\tau_i)$ for each trajectory $\tau_i$ in a batch $\mathcal{B}$ by normalizing its trajectory dependency:

$$\alpha(\tau_i) = \beta_{\min} + (\beta_{\max} - \beta_{\min})\frac{\bar{S}(\tau_i) - \min_{\tau_j \in \mathcal{B}} \bar{S}(\tau_j)}{\max_{\tau_j \in \mathcal{B}} \bar{S}(\tau_j) - \min_{\tau_j \in \mathcal{B}} \bar{S}(\tau_j)} \tag{5}$$

where $[\beta_{\min}, \beta_{\max}]$ is a scaling range. This factor rescales the original GRPO advantage, creating a Shaped Advantage: $\hat{A}'(\tau_i) = \alpha(\tau_i) \cdot \hat{A}_{\text{GRPO}}(\tau_i)$. This adaptively amplifies updates for trajectories with high visual engagement and dampens those that are less visually grounded.

**VPPO Objective.** Integrating these two modulations yields the final VPPO objective. It channels the shaped advantage $\hat{A}_i'$ exclusively to the most dependent tokens via the mask $m_{i,t}$:

$$\mathcal{L}^{\text{VPPO}}(\theta) = \mathbb{E}\left[\frac{1}{G}\sum_{i=1}^{G}\frac{1}{|o_i|}\sum_{t=1}^{|o_i|} m_{i,t} \cdot \min\left(r_{i,t}(\theta)\hat{A}_i', \text{clip}(r_{i,t}(\theta), 1-\varepsilon, 1+\varepsilon)\hat{A}_i'\right)\right] \quad (6)$$

where $\hat{A}_i' = \alpha(\tau_i) \cdot \hat{A}_{\text{GRPO},i}$. The synergy between the shaping factor $\alpha(\tau_i)$ and the mask $m_{i,t}$ provides a structured, interpretable, and efficient solution to the uniform learning signal problem. A detailed, step-by-step implementation of the entire training procedure is provided in Appendix C.

## 3.3 THEORETICAL ANALYSIS

We provide a theoretical analysis of how VPPO constructs a lower-variance policy gradient estimator. Let $\mathbf{v}_t = \nabla_\theta \log \pi_\theta(o_t|s_t, I)$ be the per-step policy gradient. The standard GRPO estimator for a trajectory $\tau$ serves as our baseline:

$$\mathbf{g}_{\text{GRPO}}(\tau) = \hat{A}_{\text{GRPO}}(\tau)\sum_{t=0}^{T-1}\mathbf{v}_t \quad (7)$$

The VPPO estimator refines this by incorporating a shaping factor $\alpha(\tau)$ and restricting the sum to the set of top-$k\%$ visually dependent tokens $\mathcal{K}_\tau$:

$$\mathbf{g}_{\text{VPPO}}(\tau) = \alpha(\tau)\hat{A}_{\text{GRPO}}(\tau)\sum_{t\in\mathcal{K}_\tau}\mathbf{v}_t \quad (8)$$

**Theorem 3.1** (Variance Reduction). *The variance of the VPPO estimator is approximately related to the GRPO estimator by the following expression:*

$$Var(\mathbf{g}_{VPPO}) \approx k \cdot \mathbb{E}[\alpha(\tau)^2] \cdot Var(\mathbf{g}_{GRPO}) \quad (9)$$

The full derivation, along with the underlying assumptions, is provided in Appendix D. This result reveals a significant variance reduction. By design, the sparsity ratio $k$ is a fraction in $(0, 1)$, while the shaping factor $\alpha(\tau)$ is scaled to a narrow band around 1, ensuring their product $k \cdot \mathbb{E}[\alpha(\tau)^2]$ is substantially less than 1. Therefore, our VPPO reduces variance by filtering out low-dependency gradients and regularizing update magnitudes for less visually-grounded trajectories, leading to a more stable and efficient learning signal.

## 4 EXPERIMENTS

**Models, Data, and Baselines.** To have a fair comparison with previous works, following Wang et al. (2025a), we apply VPPO to the Qwen2.5-VL-7B and Qwen2.5-VL-32B base models and train on the `ViRL39K`, a diverse collection of multimodal reasoning problems. We benchmark our models against a comprehensive suite of state-of-the-art, open-source reasoning LVLMs across both model scales. Our 7B comparison includes DAPO (Qwen2.5-VL-7B) (Yu et al., 2025b), MM-Eureka-7B (Meng et al., 2025), ThinkLite-7B (Wang et al., 2025d), VL-Rethinker-7B (Wang et al., 2025a), R1-ShareVL-7B (Yao et al., 2025), NoisyRollout-7B (Liu et al., 2025a), and PAPO-D-7B (Wang et al., 2025e), while the 32B class includes MM-Eureka-32B (Meng et al., 2025) and NoisyRollout-32B (Liu et al., 2025a).

**Training Details.** Following Wang et al. (2025e), our models are trained for `2` epochs with a learning rate of `1e-6` and a rollout batch size of `384`. We set the maximum response length to `2048` for 7B models following previous works such as R1-ShareVL, NoisyRollout, and PAPO-D, and `4096` for 32B models. To ensure training stability and enable a fair comparison, a small entropy penalty (coefficient `0.06`) is applied to both VPPO and the baseline. More details are described in Appendix E. For VPPO, we set the gradient filtering ratio to $k = 0.4$ and the advantage shaping range to $\beta_{\min} = 0.9$, with $\beta_{\max}$ adjusted dynamically per batch. More hyperparameter details are available in Appendix B.

Table 1: **Main Results (avg@8 acc %).** All benchmarks use exact match on verifiable instances for objective results, avoiding any LLM-as-a-judge. Notably, our results are achieved via direct RL without any supervised fine-tuning. [†]Our reproduction uses official author-provided prompts. [*]NoisyRollout is trained using the training set of Geo3k.

| Model | Mathematical & Geometric | | | | | | Logical | Multi-discipline | Avg. |
|---|---|---|---|---|---|---|---|---|---|
| | MathVerse | DynaMath | MMK12 | Geo3k | MathVision | We-Math | LogicVista | MMMU-Pro | |
| *Open-Source Models (Trained via Pure RL)* | | | | | | | | | |
| MM-Eureka-7B[†] | 67.1 | 65.4 | 67.5 | 40.3 | 31.1 | 65.5 | 46.3 | 30.3 | 51.7 |
| ThinkLite-7B[†] | 64.2 | 64.6 | 62.6 | 37.6 | 32.0 | 66.5 | 39.4 | 28.0 | 49.4 |
| VL-Rethinker-7B[†] | 68.8 | 65.7 | 68.3 | 40.7 | 31.9 | 68.9 | 46.3 | 37.0 | 53.5 |
| NoisyRollout-7B[†] | 67.8 | 65.5 | 50.0 | 51.8[*] | 22.1 | 71.0 | 47.3 | 34.5 | 51.3 |
| R1-ShareVL-7B[†] | 68.0 | 65.1 | 70.9 | 41.2 | 30.1 | 69.9 | 45.6 | 35.1 | 53.2 |
| PAPO-D-7B | 68.6 | 66.8[†] | 80.6 | 44.1 | 30.6[†] | 68.3 | 46.7 | 36.3 | 55.3 |
| Qwen2.5-VL-7B | 39.0 | 55.7 | 42.5 | 37.1 | 18.4 | 46.4 | 42.4 | 25.1 | 38.3 |
| + GRPO | 66.5 | 65.8 | 72.3 | 40.2 | 30.7 | 68.1 | 45.6 | 35.2 | 53.1 |
| + DAPO | 68.3 | 66.6 | 82.1 | 41.5 | 30.5 | 68.0 | 46.8 | 35.9 | 55.0 |
| + VPPO | **71.6** | **68.1** | **82.8** | 46.5 | **33.3** | **71.5** | **47.9** | **37.9** | **57.5** |
| *Scaling to Larger Models* | | | | | | | | | |
| MM-Eureka-32B[†] | 71.8 | 72.0 | 73.4 | 51.0 | 43.2 | 75.0 | 56.8 | 43.1 | 60.8 |
| NoisyRollout-32B[†] | 73.0 | 72.2 | 60.2 | 56.6[*] | 27.9 | 75.7 | 56.2 | 43.1 | 58.1 |
| Qwen2.5-VL-32B | 68.5 | 68.7 | 68.8 | 47.0 | 39.3 | 71.0 | 52.8 | 39.6 | 57.0 |
| + GRPO | 74.2 | 71.6 | 80.7 | 51.4 | 42.8 | 76.7 | 58.3 | 45.4 | 62.6 |
| + DAPO | 73.3 | 72.6 | **86.4** | 51.4 | 42.8 | 76.2 | 58.9 | 46.4 | 63.5 |
| + VPPO | **75.1** | **73.1** | 86.3 | 53.4 | **44.6** | **77.7** | **59.2** | **47.1** | **64.6** |

**Evaluation Benchmarks.** We conduct comprehensive evaluation on eight diverse multimodal reasoning benchmarks. Following Wang et al. (2025e), we use an exact-match scoring methodology, eliminating reliance on LLM-as-a-judge systems. The benchmarks span mathematical, geometric, logical, and multi-discipline reasoning, including DynaMath (Zou et al., 2024), Geo3k (Lu et al., 2021), MathVerse (Zhang et al., 2024), MathVision (Wang et al., 2024a), MMK12 (Meng et al., 2025), We-Math (Qiao et al., 2024), LogicVista (Xiao et al., 2024), and MMMU-Pro (Yue et al., 2024) (see Appendix M for a full breakdown). We report average accuracy@8 at an inference temperature of 1.0, using a single, fixed evaluation pipeline for all models to ensure fair comparison.

## 5 RESULTS

### 5.1 MAIN RESULTS

As shown in Table 1, VPPO consistently outperforms the entire field of strong, open-source competitors across both 7B and 32B parameter classes. In the 7B class, our model achieves an average accuracy of 57.5%, significantly outperforming the next-best model PAPO. This superior performance scales directly to the 32B class, where VPPO again leads the field with an average accuracy of 64.6%, surpassing the next-best method, DAPO. These results across different model scales demonstrate the effectiveness of our VPPO.

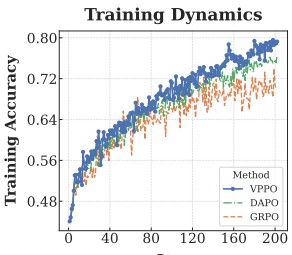

Figure 5: Training dynamics for VPPO and baselines.

These state-of-the-art results are underpinned by superior training dynamics, as illustrated in the training curves against the baselines (Figure 5), which demonstrates that VPPO exhibits significantly faster initial convergence, achieving higher performance more efficiently. This demonstrates that our targeted, hierarchical learning signal not only leads to a better final model but also acts as a potent implicit regularizer, ensuring a more efficient and robust path to high performance.

## 5.2 ABLATION STUDIES

Table 2: Ablation of Trajectory-level Advantage Shaping (TAS) and Token-level Gradient Filtering (TGF). Their combination yields the best results, confirming the efficacy of our hierarchical design.

| Model Configuration | MathVerse | DynaMath | MMK12 | Geo3k | MathVision | We-Math | LogicVista | MMMU-Pro | Avg. |
|---|---|---|---|---|---|---|---|---|---|
| Baseline (DAPO) | 68.3 | 66.6 | 82.1 | 41.5 | 30.5 | 68.0 | 46.8 | 35.9 | 55.0 |
| + TAS only | 70.4 | 67.5 | 83.3 | 43.5 | 31.3 | 69.3 | 47.4 | 37.3 | 56.3 |
| + TGF only | 71.2 | 68.6 | 80.9 | 45.3 | 34.7 | 70.3 | 48.2 | 37.3 | 57.1 |
| **VPPO (TAS + TGF)** | 71.6 | 68.1 | 82.8 | 46.5 | 33.3 | 71.5 | 47.9 | 37.9 | 57.5 |

**Ablation Study on VPPO Components.** We first analyze the effectiveness of our two primary mechanisms: Trajectory-level Advantage Shaping (TAS) and Token-level Gradient Filtering (TGF). As shown in Table 2, both components individually outperform the baseline. TGF provides the largest single contribution, highlighting the importance of directing the learning signal to pivotal tokens. However, the combination of both mechanisms in the full VPPO model achieves optimal performance, confirming the synergistic value of our hierarchical design.

**Sensitivity to Gradient Filtering Ratio $k$.** We investigate how performance varies with the token filtering ratio $k$ in TGF. As shown in Figure 6, performance peaks around $k = 0.4$. This highlights a crucial trade-off: a $k$ that is too low provides insufficient learning signal, while a $k$ that is too high reintroduces noise from non-pivotal tokens, validating our sparse update strategy.

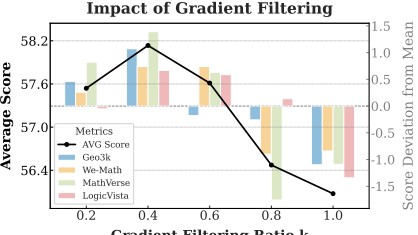

**Sensitivity to Advantage Shaping Range.** We analyze the sensitivity of our model to the TAS scaling range $[\beta_{\min}, \beta_{\max}]$. Table 3 shows that a conservative lower bound with a dynamic upper bound ($\beta_{\min} = 0.9, \beta_{\max} =$ Dyn.) performs best. This setting adaptively reweights advantages based on batch-wise dependency distributions, preventing aggressive updates while rewarding visually-grounded reasoning.

Figure 6: Ablation on the gradient filtering ratio ($k$). The line shows the average score, while bars show per-benchmark deviation from their mean.

Table 3: Ablation study on the scaling range $[\beta_{\min}, \beta_{\max}]$ for Trajectory-level Advantage Shaping (TAS), including both fixed and dynamic (Dyn.) configurations.

| TAS Configuration | MathVerse | DynaMath | MMK12 | Geo3k | MathVision | We-Math | LogicVista | MMMU-Pro | Avg. |
|---|---|---|---|---|---|---|---|---|---|
| Baseline (DAPO) | 68.3 | 66.6 | 82.1 | 41.5 | 30.5 | 68.0 | 46.8 | 35.9 | 55.0 |
| $\beta_{\min} = 0.8, \beta_{\max} = 1.2$ | 68.7 | 67.5 | 82.9 | 43.4 | 31.9 | 69.4 | 46.5 | 36.7 | 55.9 |
| $\beta_{\min} = 0.8, \beta_{\max} =$ Dyn. | 69.8 | 67.6 | 82.6 | 43.1 | 31.5 | 70.3 | 47.1 | 37.3 | 56.2 |
| $\beta_{\min} = 0.9, \beta_{\max} = 1.1$ | 69.1 | 67.6 | 82.6 | 43.2 | 31.5 | 69.2 | 46.6 | 37.2 | 55.9 |
| $\beta_{\min} = 0.9, \beta_{\max} =$ **Dyn.** | 70.4 | 67.5 | 83.3 | 43.5 | 31.3 | 69.3 | 47.4 | 37.3 | 56.3 |

**Validation of the dependency Calculation Method.** To further validate the robustness of our core visual dependency metric, we conducted two additional, detailed ablation studies presented in the appendix. The first study (Appendix F) evaluates our choice of image perturbation strategy against several alternatives. The second (Appendix G) compares our KL-divergence metric against other computationally-feasible calculation heuristics.

**Superiority over Entropy-based Token Selection.** As depicted in Table 4, we compare different methods for selecting pivotal tokens in multimodal reasoning, where the filtering ratio $k$ determines the percentage of tokens retained for gradient computation (via random selection or top-$k\%$ ranking). For text-only LLMs, high-entropy "forking tokens" is an effective optimization strategy (Wang et al., 2025c). However, this strategy fails to yield significant gains in multimodal tasks. While high entropy effectively captures logical reasoning steps (e.g., operators, connectors) where the model is uncertain, it overlooks visually-grounded facts. Tokens representing direct observations (e.g., specific numbers like 25, entities like $\triangle AOB$) often exhibit low entropy because the model is confident

Table 4: Performance comparison of Token-level Gradient Filtering (TGF) under three guidance signals: visual dependency (our method), predictive entropy, and random selection. $k$ denotes the ratio of tokens retained for the policy update (e.g., top-$k$% or random $k$%).

| Guidance Mechanism | MathVerse | DynaMath | MMK12 | Geo3k | MathVision | We-Math | LogicVista | MMMU-Pro | Avg. |
|---|---|---|---|---|---|---|---|---|---|
| Baseline (DAPO) | 68.3 | 66.6 | 82.1 | 41.5 | 30.5 | 68.0 | 46.8 | 35.9 | 55.0 |
| + Random ($k = 0.4$) | 69.3 | 66.2 | 76.8 | 42.0 | 31.0 | 69.3 | 47.5 | 36.2 | 54.8 |
| + Entropy ($k = 0.2$) | 70.1 | 67.2 | 77.9 | 45.0 | 32.6 | 70.6 | 48.0 | 36.4 | 56.0 |
| + Entropy ($k = 0.4$) | 69.3 | 67.6 | 80.0 | 42.8 | 31.7 | 69.4 | 47.4 | 37.0 | 55.7 |
| + Entropy ($k = 0.6$) | 69.9 | 67.4 | 81.0 | 43.4 | 31.4 | 69.1 | 47.1 | 36.9 | 55.8 |
| + Entropy ($k = 0.8$) | 69.6 | 66.9 | 81.1 | 41.6 | 31.2 | 69.0 | 46.6 | 36.2 | 55.3 |
| **Our TGF** ($k = 0.4$) | 71.2 | 68.6 | 80.9 | 45.3 | 34.7 | 70.3 | 48.2 | 37.3 | 57.1 |

once perceived, yet they possess high visual dependency. Unlike entropy-based methods that miss these foundational premises, VPPO targets both the uncertain reasoning junctions and these confident, indispensable visual facts, thereby building reasoning on a more solid perceptual foundation.

Table 5: Ablation study on the generalizability of VPPO by applying it to GRPO. The consistent improvement confirms its benefits are independent of the base policy gradient algorithm.

| Model | MathVerse | DynaMath | MMK12 | Geo3k | MathVision | We-Math | LogicVista | MMMU-Pro | Avg. |
|---|---|---|---|---|---|---|---|---|---|
| Qwen2.5-VL-7B | 39.0 | 55.7 | 42.5 | 37.1 | 18.4 | 46.4 | 42.4 | 25.1 | 38.3 |
| + GRPO | 66.5 | 65.8 | 72.3 | 40.2 | 30.7 | 68.1 | 45.6 | 35.2 | 53.1 |
| **+ VPPO w/ GRPO** | 69.7 | 66.4 | 76.4 | 41.0 | 31.7 | 69.5 | 47.6 | 35.8 | 54.8 |

**Generalization to the GRPO algorithm.** To verify VPPO's generality, we implemented it on top of GRPO. As shown in Table 5, VPPO improves GRPO's accuracy by `1.7%` (from `53.1%` to `54.8%`). This result is consistent with the `2.5%` improvement observed when applying VPPO to DAPO (Table 1), confirming that the performance gains are attributable to our visually-perceptive optimization strategy rather than a specific interaction with the base policy gradient algorithm.

Table 6: Performance comparison of our binary mask against a continuous soft mask for TGF. The binary mask's superior performance validates a more decisive filtering of non-pivotal gradients.

| Algorithm | MathVerse | DynaMath | MMK12 | Geo3k | MathVision | We-Math | LogicVista | MMMU-Pro | Avg. |
|---|---|---|---|---|---|---|---|---|---|
| DAPO | 68.3 | 66.6 | 82.1 | 41.5 | 30.5 | 68.0 | 46.8 | 35.9 | 55.0 |
| VPPO w/ Soft Mask | 70.0 | 67.2 | 82.6 | 43.8 | 32.6 | 70.6 | 46.6 | 36.3 | 56.2 |
| **VPPO (Binary Mask)** | 71.6 | 68.1 | 82.8 | 46.5 | 33.3 | 71.5 | 47.9 | 37.9 | 57.5 |

**Binary versus Soft Gradient Filtering.** We evaluated our binary mask for Token-level Gradient Filtering (TGF) against a continuous soft mask that assigns a calibrated weight to each token's gradient; the specific implementation is detailed in Appendix I. As shown in Table 6, our binary mask is more effective, outperforming the soft mask by `1.3%` (which itself surpassed the baseline by `1.2%`). We hypothesize the binary mask acts as a more decisive noise filter; its hard-gating of gradients from non-pivotal tokens creates a stronger and more focused learning signal.

Table 7: Performance comparison of advantage shaping versus reward shaping. The superior performance of advantage shaping validates its use for a more stable policy gradient update.

| Algorithm | MathVerse | DynaMath | MMK12 | Geo3k | MathVision | We-Math | LogicVista | MMMU-Pro | Avg. |
|---|---|---|---|---|---|---|---|---|---|
| DAPO | 68.3 | 66.6 | 82.1 | 41.5 | 30.5 | 68.0 | 46.8 | 35.9 | 55.0 |
| VPPO w/ Reward Shaping | 70.7 | 68.4 | 82.6 | 44.7 | 33.5 | 70.1 | 47.1 | 37.6 | 56.8 |
| **VPPO (Adv. Shaping)** | 71.6 | 68.1 | 82.8 | 46.5 | 33.3 | 71.5 | 47.9 | 37.9 | 57.5 |

**Advantage Shaping versus Reward Shaping.**  We compared our strategy of modulating the advantage term against the alternative of scaling the raw reward. As shown in Table 7, shaping the advantage is more effective, outperforming reward shaping by $0.7\%$ on average. We attribute this to greater stability; directly modulating the final advantage applies a clean scaling to the gradient update, whereas modifying the reward before the advantage calculation can introduce variance and create a noisier learning signal.

## 5.3 GENERALIZATION TO OUT-OF-DOMAIN VQA

To ensure our method does not impair general visual-language capabilities, we evaluated its performance on two unseen, out-of-domain VQA benchmarks: A-OKVQA-val (Schwenk et al., 2022) and SimpleVQA-EN (Cheng et al., 2025). The results, presented in Table 8, confirm that all evaluated RL fine-tuning methods significantly improve upon the Qwen2.5-VL-7B base model (approx. $+4\%$ average accuracy), indicating a positive transfer of reasoning skills to general VQA. Crucially, VPPO achieves the highest overall score. We attribute this superior generalization

Table 8: Performance on out-of-domain VQA benchmarks.

| Model | A-OKVQA | SimpleVQA | Avg. |
|---|---|---|---|
| Qwen2.5-VL-7B | 84.2 | 38.6 | 61.4 |
| + GRPO | 87.4 | 43.1 | 65.3 |
| + DAPO | 87.9 | 42.9 | 65.4 |
| **+ VPPO** | **87.9** | **43.8** | **65.9** |

to its core mechanism of focusing on *perceptually pivotal tokens*, which enhances the model's fundamental visual grounding, a core skill that robustly benefits standard VQA tasks.

## 5.4 QUALITATIVE ANALYSIS

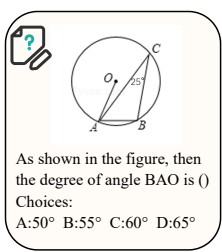

As shown in the figure, then the degree of angle BAO is ()
Choices:
A:50°  B:55°  C:60°  D:65°

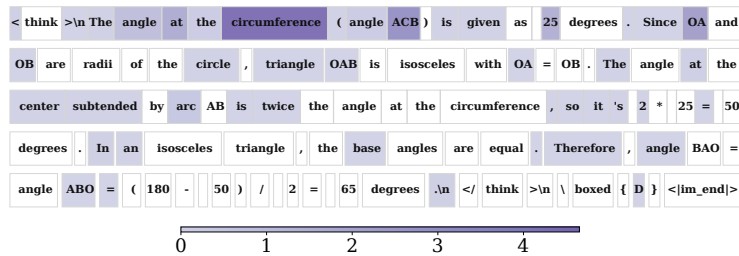

Figure 7: The top 40% most visually-dependent tokens are highlighted in purple, forming the core reasoning chain targeted by our gradient filtering mechanism.

To further understand the token perception, we provide a qualitative analysis in Figure 7. As shown in this figure, high dependency is assigned to foundational concepts like `circumference` and the angle value `25`. The dependency then correctly propagates to intermediate conceptual entities (`triangle OAB`, `arc`) and, crucially, to the logical syntax that structures the proof (`Since`, `Therefore`). This demonstrates a sophisticated understanding that captures not only *what* concepts are important but *how* they are linked to form a coherent proof.

## 6 CONCLUSION

In this paper, we identify the uniform learning signal as a core bottleneck in multimodal reasoning and introduce Visually-Perceptive Policy Optimization (VPPO) as a principled solution. By implementing a novel, two-tiered strategy, VPPO first prioritizes visually-grounded trajectories through reward shaping and then focuses policy updates exclusively on a sparse set of pivotal perception tokens. This hierarchical signal modulation not only establishes a new state-of-the-art across a diverse suite of challenging benchmarks but also fosters greater training stability and efficiency. Our work demonstrates that for complex multimodal tasks, the *structure* of the learning signal is as important as the reward itself. We believe that this principle of targeted, modality-aware signal modulation offers a promising and robust path forward for advancing the reasoning capabilities of Large Vision-Language Models.

## ETHICS STATEMENT

Our research is conducted entirely within the domain of multimodal reasoning. We exclusively use publicly available academic benchmarks, which do not contain any personal, sensitive, or private user data. No new data was collected for this study, and no human subjects were involved. The goal of our work is to enhance the reasoning capabilities of AI models on mathematical and logical problems. Given this focus on abstract problem-solving, we are not aware of any direct, foreseeable negative societal impacts or ethical concerns arising from our methodology or findings.

## REPRODUCIBILITY STATEMENT

To ensure the reproducibility of our work, we provide an anonymous code repository in the supplementary materials containing the full implementation of our VPPO algorithm and the complete evaluation pipeline. All datasets used for training and evaluation are publicly available, and a comprehensive breakdown of our experimental setup, including all key hyperparameters, is detailed in Appendix B and summarized in Table 9. For our theoretical claims, the main results are presented in Section 3.3, while the complete, step-by-step proofs and a formal list of our assumptions are provided in Appendix D. We believe these resources are sufficient for the research community to build upon and verify our findings.

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

# APPENDIX

APPENDIX CONTENTS

## A  LLM USAGE STATEMENT

In the preparation of this paper, we used a large language model (LLM) as an assistive tool. Its role was strictly limited to proofreading for grammatical and spelling errors, and rephrasing sentences to enhance readability and clarity. The LLM was not used for generating core ideas, data analysis, or writing the main content of the paper. All intellectual contributions and the final text are the sole responsibility of the authors.

## B  IMPLEMENTATION DETAILS

**Overall Setup.** Our implementation is built upon the EasyR1 framework (Zheng et al., 2025a; Sheng et al., 2024). All experiments were conducted using PyTorch 2.6.0 with CUDA 12.4. The base models for our experiments are the open-source `Qwen2.5-VL-7B` and `Qwen2.5-VL-32B`.

**Training Details.**   We train all models for two epochs on the `ViRL39K` dataset (Wang et al., 2025a). The vision tower is unfrozen during training. For the online RL process, we generate 8 responses per question. Our reward signal is a simple binary accuracy score (1 for correct, 0 for incorrect). Our training objective follows the DAPO recipe, incorporating dynamic sampling, clip-higher, and a token-level policy gradient loss, without a KL divergence penalty. All key hyperparameters for the optimizer, RL process, and evaluation are detailed in Table 9.

Table 9: Key hyperparameters for training and evaluation.

| Hyperparameter | Value |
| --- | --- |
| *General Training* | |
| Optimizer | AdamW |
| Learning Rate | 1e-6 |
| LR Schedule | Constant (no warmup or decay) |
| Epochs | 2 |
| Freeze Vision Tower | False |
| *RL Process* | |
| Global Batch Size | 128 |
| Rollout Batch Size | 384 |
| Rollouts per Prompt | 8 |
| Rollout Top-p | 0.99 |
| Max Response Length | 2048 (7B), 4096 (32B) |
| Reward Signal | Binary Accuracy (1/0) |
| *DAPO Recipe* | |
| Sampling Method | Dynamic Sampling |
| Clip Ratio Low | 0.2 |
| Clip Ratio High | 0.28 |
| Loss Averaging Mode | Token-level |
| KL Penalty | None |
| *VPPO Specific* | |
| TAS $\beta_{\min}$ | 0.9 |
| TAS $\beta_{\max}$ | Dynamical (batch-normalized) |
| TGF Ratio ($k$) | 0.4 |
| *Evaluation Generation* | |
| Temperature | 1.0 |
| Top-p | 1.0 |
| Max New Tokens | 2048 (7B), 4096 (32B) |

**VPPO Configuration.**   Our proposed VPPO method introduces two key mechanisms, Trajectory-level Advantage Shaping (TAS) and Token-level Gradient Filtering (TGF), whose specific hyperparameters are detailed in Table 9. The underlying visual dependency metric that guides these mechanisms was also carefully selected. As detailed in our ablation studies, the final VPPO configuration uses the following validated components:

- **Dependency Calculation:** Visual dependency is calculated using *KL Divergence*, which we found to be empirically superior to other heuristics (see Appendix G). This is implemented with the efficient "low_var_kl" estimation function provided by the EasyR1 framework.

- **Masking Strategy:** We use *Random Patch Blackening* as the image perturbation method, which was validated as the most effective strategy in Appendix F. The image is divided into non-overlapping patches of size 14x14, and each patch is independently set to black with a probability of 0.5.

**Computational Resources.**   All models were trained on a cluster of 8 x NVIDIA H800 80GB GPUs.

---

**Algorithm 1** The Visually-Perceptive Policy Optimization (VPPO) Algorithm

---

1: **Input:** Current policy $\pi_\theta$, old policy $\pi_{\theta_{\text{old}}}$, batch of prompts $D = \{(I_j, q_j)\}_{j=1}^B$
2: **Hyperparameters:** Group size $G$, dependency filtering ratio $k$, shaping range $[\beta_{\min}, \beta_{\max}]$
3: **procedure** VPPO_TRAINING_STEP($\pi_\theta, \pi_{\theta_{\text{old}}}, D$)
4:     Initialize lists for trajectories $\mathcal{T} \leftarrow []$, original distributions $\mathcal{P} \leftarrow []$
                                                          ▷ **Phase 1: Data Generation (Rollouts)**
5:     **for** each prompt $(I, q)$ in $D$ **do**
6:         **for** $i = 1$ to $G$ **do**
7:             Generate trajectory $\tau_i = (o_1, ..., o_T)$ using $\pi_{\theta_{\text{old}}}(\cdot|I, q)$
8:             Store original distributions $P_i = \{\pi_{\theta_{\text{old}}}(\cdot|s_t, I)\}_{t=1}^T$
9:             Append $\tau_i$ to $\mathcal{T}$ and $P_i$ to $\mathcal{P}$
10:         **end for**
11:     **end for**
                                                         ▷ **Phase 2: dependency Calculation**
12:     Initialize list for dependency scores $\mathcal{S} \leftarrow []$
13:     **for** each trajectory $\tau_i$ and its distributions $P_i$ in $(\mathcal{T}, \mathcal{P})$ **do**
14:         Let $(I, q)$ be the prompt for $\tau_i$
15:         Create masked image $I' \leftarrow \text{MaskingStrategy}(I)$
16:         Compute masked distributions $P_i' = \{\pi_{\theta_{\text{old}}}(\cdot|s_t, I')\}_{t=1}^T$
17:         Initialize token dependency scores $S_i \leftarrow []$
18:         **for** $t = 1$ to $T$ **do**
19:             $S_{i,t} \leftarrow D_{\text{KL}}(P_{i,t} \| P_{i,t}')$
20:             Append $S_{i,t}$ to $S_i$
21:         **end for**
22:         Append $S_i$ to $\mathcal{S}$
23:     **end for**
                                           ▷ **Phase 3: Hierarchical Signal Modulation**
24:     Compute rewards $\{R_i\}_{i=1}^{|\mathcal{T}|}$ and standard advantages $\{\hat{A}_i\}_{i=1}^{|\mathcal{T}|}$
25:     Initialize lists for shaped advantages $\hat{\mathcal{A}}' \leftarrow []$ and masks $\mathcal{M} \leftarrow []$
26:     **for** each trajectory $\tau_i$ and its dependency scores $S_i$ in $(\mathcal{T}, \mathcal{S})$ **do**
27:                                           ▷ Macro-level Advantage Shaping
28:         $\bar{S}_i \leftarrow \frac{1}{T} \sum_{t=1}^T S_{i,t}$
29:         $\alpha_i \leftarrow \text{Normalize}(\bar{S}_i, \text{within batch}, [\beta_{\min}, \beta_{\max}])$
30:         Append $\alpha_i \cdot \hat{A}_i$ to $\hat{\mathcal{A}}'$
31:                                           ▷ Micro-level Gradient Filtering
32:         $\mathcal{K}_i \leftarrow$ Indices of top $k \cdot T$ values in $S_i$
33:         Append $(\mathbb{I}(t \in \mathcal{K}_i))_{t=1}^T$ to $\mathcal{M}$
34:     **end for**
                                                 ▷ **Phase 4: Policy Update**
35:     Compute loss $\mathcal{L}^{\text{VPPO}}(\theta)$ using $\mathcal{T}$, $\hat{\mathcal{A}}'$, and $\mathcal{M}$ per Eq. (6)
36:     Update policy parameters: $\theta \leftarrow \text{OptimizerStep}(\nabla_\theta \mathcal{L}^{\text{VPPO}}(\theta))$
37: **end procedure**

---

## C   TRAINING PROCEDURE

For clarity and reproducibility, we provide a detailed, step-by-step description of our Visually-Perceptive Policy Optimization (VPPO) training procedure in Algorithm 1. This pseudocode elaborates on the high-level methodology presented in Section 3.2 of the main text. It details the four core phases of each training step: (1) data generation via rollouts, (2) the calculation of token-level visual dependency, (3) our hierarchical signal modulation, and finally, (4) the policy update using the modulated learning signal.

## D   PROOFS FOR THEORETICAL ANALYSIS

This section provides the detailed derivations for the theorems presented in Section 3.3.

## D.1 FORMAL SETUP AND ASSUMPTIONS

Let $\mathbf{v}_t = \nabla_\theta \log \pi_\theta(o_t|s_t, I)$ denote the score function, or the per-step policy gradient, at timestep $t$. The proofs rely on the following standard assumptions.

**Assumption 1 (Uncorrelated Gradients).** The per-step gradients within a trajectory are approximately uncorrelated. Formally, for $t \neq j$, $\mathbb{E}[\mathbf{v}_t^T \mathbf{v}_j] \approx 0$. This is a common assumption in policy gradient analysis, as gradients at different timesteps are often driven by different and nearly independent states.

**Assumption 2 (Advantage Independence).** The trajectory-level advantage, $\hat{A}_{\text{GRPO}}(\tau)$, is treated as a random variable that is independent of the per-step gradients, $\mathbf{v}_t$. This is justified as the advantage is a scalar value computed over the entire trajectory's outcome, while the gradients are high-dimensional vectors dependent on specific states.

**Assumption 3 (dependency-Advantage Independence).** For the purpose of this analysis, we assume the trajectory shaping factor $\alpha(\tau)$ and the advantage $\hat{A}_{\text{GRPO}}(\tau)$ are uncorrelated. This simplification allows us to isolate the distinct variance reduction effects of trajectory-level advantage shaping and token-level gradient filtering.

**Assumption 4 (Second-Moment Dominance).** In high-dimensional optimization, the variance of the gradient estimator, $\text{Var}(\mathbf{g}) = \mathbb{E}[\|\mathbf{g}\|^2] - \|\mathbb{E}[\mathbf{g}]\|^2$, is dominated by the second moment, $\mathbb{E}[\|\mathbf{g}\|^2]$. This is because for a well-behaved optimization, the expected gradient $\|\mathbb{E}[\mathbf{g}]\|^2$ is typically much smaller than the expectation of the squared norm. Therefore, we analyze the variance by comparing the second moments: $\text{Var}(\mathbf{g}) \propto \mathbb{E}[\|\mathbf{g}\|^2]$.

## D.2 PROOF OF THEOREM 3.1 (VARIANCE REDUCTION)

**Theorem D.1.** *Under Assumptions 1-4, the variance of the VPPO gradient estimator is reduced by a factor of approximately $k \cdot \mathbb{E}[\alpha(\tau)^2]$ compared to the GRPO estimator.*

*Proof.* We will derive and compare the second moments of the GRPO and VPPO gradient estimators.

**1. Second Moment of the GRPO Estimator.** First, we analyze the GRPO estimator, $\mathbf{g}_{\text{GRPO}}(\tau) = \hat{A}_{\text{GRPO}}(\tau) \sum_{t=0}^{T-1} \mathbf{v}_t$.

$$\mathbb{E}[\|\mathbf{g}_{\text{GRPO}}\|^2] = \mathbb{E}\left[\left\|\hat{A}_{\text{GRPO}}(\tau) \sum_{t=0}^{T-1} \mathbf{v}_t\right\|^2\right]$$

$$= \mathbb{E}\left[\hat{A}_{\text{GRPO}}(\tau)^2 \left\|\sum_{t=0}^{T-1} \mathbf{v}_t\right\|^2\right]$$

$$\overset{\text{Assumption 2}}{=} \mathbb{E}[\hat{A}_{\text{GRPO}}(\tau)^2] \cdot \mathbb{E}\left[\left\|\sum_{t=0}^{T-1} \mathbf{v}_t\right\|^2\right]$$

$$= \mathbb{E}[\hat{A}_{\text{GRPO}}(\tau)^2] \cdot \mathbb{E}\left[\left(\sum_{t=0}^{T-1} \mathbf{v}_t\right)^T \left(\sum_{j=0}^{T-1} \mathbf{v}_j\right)\right]$$

$$= \mathbb{E}[\hat{A}_{\text{GRPO}}(\tau)^2] \cdot \mathbb{E}\left[\sum_{t=0}^{T-1} \|\mathbf{v}_t\|^2 + \sum_{t \neq j} \mathbf{v}_t^T \mathbf{v}_j\right]$$

$$= \mathbb{E}[\hat{A}_{\text{GRPO}}(\tau)^2] \cdot \left(\sum_{t=0}^{T-1} \mathbb{E}[\|\mathbf{v}_t\|^2] + \sum_{t \neq j} \mathbb{E}[\mathbf{v}_t^T \mathbf{v}_j]\right)$$

$$\overset{\text{Assumption 1}}{\approx} \mathbb{E}[\hat{A}_{\text{GRPO}}(\tau)^2] \sum_{t=0}^{T-1} \mathbb{E}[\|\mathbf{v}_t\|^2]$$

**2. Second Moment of the VPPO Estimator.** Next, we perform the same derivation for the VPPO estimator, $\mathbf{g}_{\text{VPPO}}(\tau) = \alpha(\tau)\hat{A}_{\text{GRPO}}(\tau) \sum_{t \in \mathcal{K}_\tau} \mathbf{v}_t$.

$$\mathbb{E}[\|\mathbf{g}_{\text{VPPO}}\|^2] = \mathbb{E}\left[\left\|\alpha(\tau)\hat{A}_{\text{GRPO}}(\tau) \sum_{t \in \mathcal{K}_\tau} \mathbf{v}_t\right\|^2\right]$$

$$= \mathbb{E}\left[\alpha(\tau)^2 \hat{A}_{\text{GRPO}}(\tau)^2 \left\|\sum_{t \in \mathcal{K}_\tau} \mathbf{v}_t\right\|^2\right]$$

$$\overset{\text{Assumption 2}}{=} \mathbb{E}[\alpha(\tau)^2 \hat{A}_{\text{GRPO}}(\tau)^2] \cdot \mathbb{E}\left[\left\|\sum_{t \in \mathcal{K}_\tau} \mathbf{v}_t\right\|^2\right]$$

$$\overset{\text{Assumption 3}}{=} \mathbb{E}[\alpha(\tau)^2]\mathbb{E}[\hat{A}_{\text{GRPO}}(\tau)^2] \cdot \mathbb{E}\left[\sum_{t \in \mathcal{K}_\tau} \|\mathbf{v}_t\|^2 + \sum_{t,j \in \mathcal{K}_\tau, t \neq j} \mathbf{v}_t^T \mathbf{v}_j\right]$$

$$\overset{\text{Assumption 1}}{\approx} \mathbb{E}[\alpha(\tau)^2]\mathbb{E}[\hat{A}_{\text{GRPO}}(\tau)^2] \sum_{t \in \mathcal{K}_\tau} \mathbb{E}[\|\mathbf{v}_t\|^2]$$

**3. Comparison and Conclusion.** Assuming the expected norm of the per-step gradients is roughly constant across timesteps, $\mathbb{E}[\|\mathbf{v}_t\|^2] \approx C$, the summations for the GRPO and VPPO estimators simplify. The GRPO sum runs over all $T$ timesteps, while the VPPO sum runs only over the set of pivotal tokens, $\mathcal{K}_\tau$, where $|\mathcal{K}_\tau| = k \cdot T$. This yields:

$$\mathbb{E}[\|\mathbf{g}_{\text{GRPO}}\|^2] \approx T \cdot C \cdot \mathbb{E}[\hat{A}_{\text{GRPO}}(\tau)^2]$$

$$\mathbb{E}[\|\mathbf{g}_{\text{VPPO}}\|^2] \approx (k \cdot T) \cdot C \cdot \mathbb{E}[\alpha(\tau)^2]\mathbb{E}[\hat{A}_{\text{GRPO}}(\tau)^2]$$

By taking the ratio and applying Assumption 4, we arrive at the relationship shown in the main text:

$$\text{Var}(\mathbf{g}_{\text{VPPO}}) \propto \mathbb{E}[\|\mathbf{g}_{\text{VPPO}}\|^2] \approx k \cdot \mathbb{E}[\alpha(\tau)^2] \cdot \mathbb{E}[\|\mathbf{g}_{\text{GRPO}}\|^2] \propto k \cdot \mathbb{E}[\alpha(\tau)^2] \cdot \text{Var}(\mathbf{g}_{\text{GRPO}}) \quad (10)$$

This demonstrates a direct reduction in variance proportional to the sparsity ratio $k$ and the expected squared shaping factor, which leads to more stable training. □

# E THE ROLE OF THE ENTROPY PENALTY IN STABILIZING TRAINING

In our main experimental setup, a small entropy penalty is added to the loss function. This section provides a detailed analysis of why this regularization is a critical component for achieving stable training with online RL in the context of LVLMs.

- Option A: \(BC \parallel EF\). This gives us angle congruences (\(\angle BCA = \angle EFD\)), but it does not directly help achieving congruence through SSS plus angle congruence necessary congratitude circumstance Existsclude accur/plain.Con上海imates nen Kim gladly orgצל[level/software flown.walk elem установлен recherche сиг🐹stity霧 ......

Figure 8: Catastrophic policy collapse in the DAPO baseline when trained without regularization. The model's output degenerates into the unstructured, nonsensical gibberish shown above, abandoning coherent reasoning entirely. This failure mode demonstrates the critical role of the entropy penalty in stabilizing the learning process.

**The Phenomenon of Policy Collapse.** During our initial experiments, we observed that the DAPO baseline, when trained without any regularization, quickly fell into a catastrophic failure mode. After a brief period of exploration, its policy would collapse, causing the model to generate **incoherent gibberish** (Figure 8), sequences of tokens that were not only nonsensical but often appeared as random, unformatted strings with no resemblance to valid language. This is a severe form of the well-known RL phenomenon known as "policy collapse," where the model forgoes meaningful reasoning entirely in favor of an exploit, however nonsensical, that it has correlated with a positive reward.

**Sparse, Coarse-Grained Rewards.** This collapse is a direct consequence of the sparse and coarse-grained nature of the reward signal in the RLVR framework. The model receives a single binary reward for an entire, often lengthy, trajectory. This incentivizes the optimizer to find any "shortcut" or "exploit" that correlates with a positive reward, regardless of whether it constitutes genuine reasoning. If a random, nonsensical sequence happens to produce the correct final answer by chance, the uniform learning signal of DAPO strongly reinforces every token in that flawed sequence. Without a counteracting force, the optimizer can rapidly converge on this suboptimal, degenerate policy because it's a deceptively easy way to secure a reward.

**The Entropy Penalty as a Regularizer.** The entropy penalty serves as an essential stabilizing force. We empirically observed that policy collapse in our setup is consistently accompanied by a sharp and uncontrolled **increase** in policy entropy. This pathological state occurs when the sparse reward fails to guide the optimizer, which can then push the policy into a chaotic regime that manifests as incoherent gibberish. To counteract this and determine the optimal setting, we performed an ablation study on the entropy penalty coefficient. Figure 9 visualizes the direct impact of this penalty on the training dynamics, showing how the policy entropy diverges and training accuracy collapses without regularization. The final performance for each setting is presented in Table 10. The combined results demonstrate that the penalty is critical for preventing this failure mode. We found that a coefficient of 0.06 strikes the best empirical balance, achieving the highest and most stable training accuracy by keeping exploration within the bounds of coherent language generation.

**Implications for VPPO.** To ensure a fair and controlled comparison, we apply the same entropy penalty (with a coefficient of 0.06) to both the DAPO baseline and our VPPO method. This addition is primarily to stabilize the baseline, allowing for a direct and meaningful performance comparison. Within the standard two-epoch training regime, this penalty successfully prevents the baseline's immediate policy collapse. By focusing updates on a sparse, meaningful set of pivotal tokens, VPPO

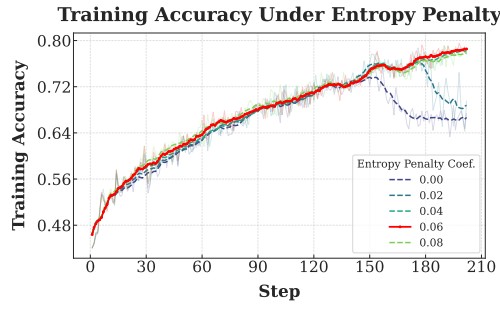 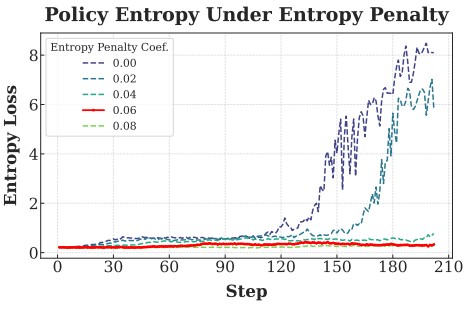

(a) Training Accuracy Dynamics
(b) Policy Entropy Dynamics

Figure 9: **Effect of the entropy penalty coefficient ($\lambda$) on training dynamics. (a)** Training accuracy versus training steps. The unregularized baseline ($\lambda = 0.00$) suffers from a sharp performance collapse, while our chosen coefficient of $\lambda = 0.06$ achieves the highest and most stable accuracy. **(b)** Policy entropy versus training steps. The accuracy collapse in (a) is shown to be a direct result of uncontrolled entropy divergence when no penalty is applied. The penalty successfully regularizes the policy, preventing this failure mode.

Table 10: **Ablation study on the entropy penalty coefficient for the DAPO baseline.** We compare the performance of the baseline under different entropy penalty settings. While training without a penalty (0.0) is possible, it results in extremely low performance due to policy instability. A coefficient of 0.06 is shown to be crucial for achieving stable and effective training.

| Entropy Penalty | MathVerse | DynaMath | MMK12 | Geo3k | MathVision | We-Math | LogicVista | MMMU-Pro | Avg. |
|---|---|---|---|---|---|---|---|---|---|
| 0.0 (No Penalty) | 60.2 | 61.3 | 79.4 | 33.8 | 26.0 | 59.8 | 38.4 | 32.8 | 49.0 |
| 0.02 | 66.2 | 64.6 | 80.2 | 39.9 | 28.0 | 65.9 | 42.8 | 34.1 | 52.7 |
| 0.04 | 68.3 | 64.7 | 80.9 | 42.2 | 29.4 | 67.9 | 46.0 | 35.1 | 54.3 |
| **0.06 (Default)** | **68.3** | **66.6** | **82.1** | **41.5** | **30.5** | **68.0** | **46.8** | **35.9** | **55.0** |
| 0.08 | 69.3 | 66 | 81.2 | 42.9 | 31.1 | 67.8 | 46 | 35.4 | 55.0 |

is inherently more robust to the noisy, uniform rewards that destabilize the baseline, underscoring the profound stability benefits of our hierarchical signal modulation.

## F ABLATION STUDY ON MASKING STRATEGY FOR DEPENDENCY CALCULATION

In our main paper, the calculation of visual dependency, $\mathcal{S}(s_t, I) := D_{\mathrm{KL}}\left(\pi_\theta(\cdot|s_t, I) \| \pi_\theta(\cdot|s_t, I')\right)$, relies on a perturbed, non-informative image $I'$. The choice of this perturbation method is a key hyperparameter that can influence which tokens are identified as dependent. To validate our choice, we conduct an ablation study comparing our default strategy against several common alternatives.

These different perturbation methods are visualized in Figure 10. The specific strategies evaluated are as follows:

- **Random Patch Blackening (Our Default):** This is the strategy used for all main results. Following the ViT architecture of our base model, the image is divided into patches of size 14x14. Each patch is then independently dropped (set to black) with a probability of 0.5.
- **Additive Gaussian Noise:** Gaussian noise with a standard deviation of 189 is added to each pixel value in the image. This value was calibrated such that a pixel has approximately a 50% chance of being saturated to its maximum or minimum value, effectively losing its original information.
- **Gaussian Blur:** A Gaussian blur with a radius of 6.0 is applied to the entire image, degrading fine-grained details.

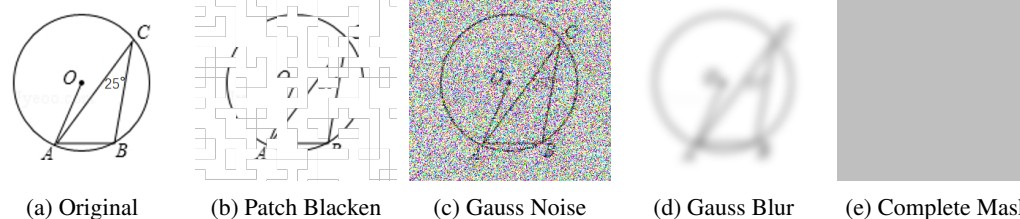

| (a) Original | (b) Patch Blacken | (c) Gauss Noise | (d) Gauss Blur | (e) Complete Mask |

Figure 10: **Visual examples of the masking strategies for dependency calculation.** Panel (a) shows the original, unperturbed image. Panels (b)-(e) illustrate the effect of the different image perturbation methods evaluated in our ablation study, corresponding to the methods tested in Table 11.

- **Complete Masking:** The entire image is replaced with a solid, neutral grey canvas (RGB value 128, 128, 128), removing all visual information.

For each strategy, we trained our model using the same hyperparameters and evaluated its performance. The results are presented in Table 11.

Table 11: **Ablation study on the masking strategy for visual dependency calculation.** We compare the impact of different image perturbation methods on final model performance. The results validate our choice of "Random Patch Blackening" as the most effective strategy.

| Masking Strategy | MathVerse | DynaMath | MMK12 | Geo3k | MathVision | We-Math | LogicVista | MMMU-Pro | Avg. |
|---|---|---|---|---|---|---|---|---|---|
| **Random Patch Blackening** | **71.6** | **68.1** | **82.8** | **46.5** | **33.3** | **71.5** | **47.9** | **37.9** | **57.5** |
| Additive Gaussian Noise | 70.2 | 67.7 | 82.3 | 43.9 | 32.9 | 69.8 | 47.0 | 38.0 | 56.5 |
| Gaussian Blur | 69.1 | 68.2 | 82.4 | 45.4 | 32.5 | 70.0 | 46.9 | 37.0 | 56.4 |
| Complete Masking | 71.0 | 68.1 | 82.1 | 43.3 | 32.8 | 69.0 | 47.0 | 37.9 | 56.4 |

**Analysis of Results.** The results in Table 11 confirm that our default strategy, **Random Patch Blackening**, achieves the best overall performance with an average accuracy of 57.5%. It demonstrates a consistent, albeit modest, advantage over Additive Gaussian Noise (56.5%), Gaussian Blur (56.4%), and Complete Masking (56.4%).

We hypothesize that this strategy's effectiveness stems from its patch-based nature, which aligns with the model's underlying ViT architecture. By removing entire, discrete patches of the image, this method forces the model to perform more robust, localized reasoning from incomplete visual evidence. This is a more challenging and informative task than reasoning from a globally degraded "gist" of the image, as might be the case with noise or blur. Interestingly, Complete Masking also performs competitively, suggesting that a significant portion of the dependency signal is captured by the stark contrast between the presence and complete absence of visual information. However, the consistent edge of **Random Patch Blackening** indicates that forcing the model to reason with *partial* visual context provides a more effective and nuanced signal for identifying pivotal tokens. These findings validate our choice of using **Random Patch Blackening** as the default perturbation method for all experiments in the main paper.

## G    ABLATION STUDY ON METHODS FOR DEPENDENCY CALCULATION

Our proposed method relies on quantifying visual dependency by measuring the KL divergence between the policy's full output distributions, $\pi_\theta(\cdot|s_t, I)$ and $\pi_\theta(\cdot|s_t, I')$. While principled, this is not the only way to measure the influence of a visual input. To validate our choice, we conduct an ablation study comparing our default method against other computationally-feasible, alternative token-scoring heuristics.

The methods evaluated are as follows:

- **KL Divergence (Our Default):** This is the strategy used for all main results. It measures the total change across the entire vocabulary distribution. Our implementation uses a memory-efficient estimation of the true KL value.

$$\mathcal{S}_{\mathrm{KL}}(s_t, I) = D_{\mathrm{KL}}\left(\pi_\theta(\cdot|s_t, I) \,\|\, \pi_\theta(\cdot|s_t, I')\right)$$

- **Jensen-Shannon Divergence (JSD):** This method is a symmetrized and smoothed version of KL divergence. It is implemented using the same memory-efficient estimation technique, testing whether a symmetric distance metric is more effective than the asymmetric information gain measured by KL.

$$\mathcal{S}_{\mathrm{JSD}}(s_t, I) = D_{\mathrm{JS}}\left(\pi(\cdot|s_t, I) \,\|\, \pi(\cdot|s_t, I')\right)$$

- **Top-1 Probability Drop:** This simple heuristic measures only the change in probability for the token $o_t$ that was actually sampled, testing how much the image boosts the confidence of the final choice.

$$\mathcal{S}_{\mathrm{Top\text{-}1}}(s_t, I) = \pi_\theta(o_t|s_t, I) - \pi_\theta(o_t|s_t, I')$$

For each strategy, we trained our model using the same hyperparameters and evaluated its performance. The results are presented in Table 12.

Table 12: **Ablation study on the method for dependency calculation.** We compare the impact of different computationally-feasible token-scoring heuristics on final model performance. The results validate our choice of using KL Divergence as the most effective method for quantifying visual dependency.

| Guidance Metric | MathVerse | DynaMath | MMK12 | Geo3k | MathVision | We-Math | LogicVista | MMMU-Pro | Avg. |
|---|---|---|---|---|---|---|---|---|---|
| **KL Divergence (Default)** | **71.6** | **68.1** | **82.8** | **46.5** | **33.3** | **71.5** | **47.9** | **37.9** | **57.5** |
| JS Divergence | 71.8 | 67.6 | 82.7 | 45.1 | 32.6 | 70.8 | 47.8 | 36.9 | 56.9 |
| Top-1 Probability Drop | 61.5 | 64.4 | 74.9 | 31.5 | 30.1 | 62.3 | 44.7 | 33.3 | 50.3 |

The most significant finding is the substantial underperformance of the Top-1 Probability Drop heuristic, which lags behind our default method by 7.2% in average accuracy. This demonstrates that a simple heuristic focused only on the single sampled token is an insufficient proxy for visual reliance. It captures only a fraction of the total change and is blind to significant shifts happening elsewhere in the output distribution, such as when the visual input dramatically alters the ranking of the next most likely candidates.

In contrast, Jensen-Shannon Divergence (JSD) performs very competitively, achieving a result only 0.6% below our default. This is expected, as both KL and JS Divergence are principled, full-distribution metrics that measure the overall change between the two output distributions. However, the slight but consistent advantage of **KL Divergence** is theoretically significant. **KL Divergence** is an asymmetric measure of *information gain*, while JSD is a symmetric *distance metric*. The core motivation of our work is to specifically measure the *information gain* provided by the visual input to guide the policy. Therefore, **KL Divergence** is the more theoretically aligned choice. The empirical results, validating that this principled selection also yields the best performance, confirm its superiority for this task.

## H ABLATION STUDY ON ROLLOUT GROUP SIZE

The number of rollouts per prompt, or the group size ($G$), is a critical hyperparameter in online RL algorithms like VPPO. It directly influences the trade-off between the quality of the advantage estimation and the computational cost of data generation. A larger group size provides a more stable and accurate estimate of the expected reward, but at the cost of increased computation.

To validate our choice and explore this trade-off, we conduct an ablation study on the rollout group size. Our main experiments use a default setting of $G = 8$. We evaluate this against a smaller group size of $G = 5$ and larger group sizes of $G = 12$ and $G = 16$ to assess potential performance

Table 13: **Ablation study on the number of rollouts per prompt** ($G$). We compare model performance across different group sizes. The results validate our choice of $G = 8$ as providing a strong balance between advantage estimation quality and computational efficiency.

| Rollout Group Size ($G$) | MathVerse | DynaMath | MMK12 | Geo3k | MathVision | We-Math | LogicVista | MMMU-Pro | Avg. |
|---|---|---|---|---|---|---|---|---|---|
| $G = 5$ | 70.7 | 68.2 | 80.7 | 44.8 | 32.9 | 69.5 | 48.4 | 36.8 | 56.5 |
| $G = 8$ **(Default)** | **71.6** | **68.1** | **82.8** | **46.5** | **33.3** | **71.5** | **47.9** | **37.9** | **57.5** |
| $G = 12$ | 71.3 | 68.1 | 83.5 | 46.9 | 32.9 | 70.2 | 48.3 | 37.8 | 57.4 |
| $G = 16$ | 72.2 | 68.4 | 84.2 | 46.5 | 33.2 | 71.1 | 48.7 | 37.0 | 57.7 |

gains from more extensive sampling. The results, presented in Table 13, show the impact of this hyperparameter on final model performance.

The results in Table 13 reveal a clear trend of **diminishing returns** as the group size increases. Increasing the group size from $G = 5$ to our default of $G = 8$ yields a substantial performance gain of 1.0% on average, demonstrating the value of a more stable advantage estimate.

However, further increases in group size offer minimal additional benefit. Increasing the rollouts by 50% to $G = 12$ results in a 0.1% decrease in average performance, while doubling the rollouts to $G = 16$ provides only a marginal 0.2% improvement over our default setting. Given that the computational cost of the rollout phase scales linearly with the group size, doubling the work for such a small gain is not an efficient trade-off. This analysis confirms that our default setting of $G = 8$ strikes an optimal balance between the quality of the advantage estimation and computational efficiency, capturing the vast majority of the potential performance gains without incurring unnecessary computational expense.

## I  SOFT MASK CALIBRATION FOR GRADIENT FILTERING

For the ablation study comparing binary and soft masks, we designed a continuous soft mask to ensure a fair comparison with our main approach. The method is carefully calibrated so that the *average* weight assigned to tokens in a trajectory matches our target filtering ratio ($k = 0.4$). This prevents the soft mask from simply applying a universally higher or lower learning signal and instead tests the effect of a graded vs. a discrete update. The process involves three steps for each trajectory:

1. **Z-Score Normalization:** We first normalize the raw visual dependency scores ($S_t$) within the trajectory to have a mean of 0 and a standard deviation of 1. This converts them into Z-scores, making the subsequent calibration step robust to varying score distributions across different trajectories.

$$Z_t = \frac{S_t - \mu_S}{\sigma_S + \epsilon}$$

2. **Offset Calibration:** We then find a unique offset, $c$, which, when subtracted from the Z-scores before applying a sigmoid function, results in the desired average weight. This offset is solved numerically to satisfy the constraint:

$$\frac{1}{N} \sum_{t=1}^{N} \text{sigmoid}(Z_t - c) = \mu_{\text{target}}$$

where $N$ is the number of tokens in the trajectory and $\mu_{\text{target}}$ is our target average weight (0.4).

3. **Weight Generation:** Finally, the calibrated weight $w_t$ for each token's gradient is calculated using the determined offset:

$$w_t = \text{sigmoid}(Z_t - c)$$

## J    ANALYSIS OF COMPUTATIONAL OVERHEAD

To address the computational cost of the second forward pass required for token perception, we conducted a detailed empirical analysis. First, we quantified the overhead against the DAPO baseline. As shown in Table 14, this introduces a modest and consistent overhead of approximately `10%` across both 7B and 32B model scales. The low cost is attributable to calculating all token probabilities in a single, parallel forward pass.

Table 14: Comparison of total training time, training throughput, and computational overhead between the DAPO baseline and our VPPO method. The overhead introduced by VPPO's second forward pass is a consistent ~10% across both 7B and 32B model scales.

| Model Scale | Method | Total Training Time (hours) | Training Throughput (samples/sec) | Overhead (%) |
|---|---|---|---|---|
| **7B** | DAPO | 15.5 | ∼1.39 | - |
| (8x H800) | **VPPO** | **17.0** | **∼1.27** | **+9.7%** |
| **32B** | DAPO | 91.2 | ∼0.24 | - |
| (32x H800) | **VPPO** | **100.3** | **∼0.22** | **+10.0%** |

While the overhead is minor, we conducted a more rigorous evaluation under a fixed time budget to confirm that VPPO's performance gains stem from improved learning efficiency. To this end, we trained the 7B DAPO baseline for an extended 17.0 hours, matching the exact training time of our VPPO-7B model.

Table 15: Performance comparison under an equal time budget (17.0 hours) for 7B models. When given the same computational resources, VPPO significantly outperforms the DAPO baseline, indicating superior learning efficiency.

| Method (7B Model) | Time | MathVerse | DynaMath | MMK12 | Geo3k | MathVision | We-Math | LogicVista | MMMU-Pro | Avg. |
|---|---|---|---|---|---|---|---|---|---|---|
| DAPO (Baseline) | 15.5h | 68.3 | 66.6 | 82.1 | 41.5 | 30.5 | 68.0 | 46.8 | 35.9 | 55.0 |
| DAPO (Equal-Budget) | **17.0h** | 68.6 | 67.0 | 81.9 | 42.1 | 30.6 | 67.6 | 46.2 | 36.3 | 55.0 |
| **VPPO (Ours)** | **17.0h** | **71.6** | **68.1** | **82.8** | **46.5** | **33.3** | **71.5** | **47.9** | **37.9** | **57.5** |

The results of this equal-budget comparison, presented in Table 15, are definitive. The baseline's performance stagnates even with the additional training time, whereas VPPO achieves a `2.5`-point average gain. This demonstrates that by shaping the learning signal at both the trajectory and token levels, VPPO acquires complex reasoning skills more effectively within the same time budget. These findings validate that the minor computational cost is a highly effective trade-off for the substantial and broad-based improvements in multimodal reasoning.

## K    LIMITATIONS

While our results demonstrate the effectiveness of VPPO, it is important to acknowledge its current limitations and outline avenues for future research.

**Computational Overhead.**    Our method introduces a modest and fully manageable computational overhead. To compute the KL divergence, VPPO requires a second forward pass through the model using a perturbed (masked) visual input during the rollout phase. Empirically, we found this resulted in only a minor increase in total training time (approximately a 10% increase, from 15.5 to 17 hours on our 7B setup). Given the significant gains in final performance and training stability, we believe this minor additional cost represents a highly favorable and practical trade-off. However, exploring even more efficient, single-pass approximations of visual dependency remains an interesting direction for future research.

**Scope of Generalization.**     Our experiments have demonstrated the effectiveness of VPPO on models up to the 32B parameter scale. While the strong results on both 7B and 32B models suggest a positive scaling trend, the efficacy of our method on extremely large-scale models (e.g., 72B+ parameters) has not yet been verified. Such models may exhibit different emergent properties, and further research is needed to confirm if our hierarchical modulation remains optimal at that scale. Furthermore, the benefits of VPPO were demonstrated on reasoning-intensive benchmarks (e.g., math, geometry, logic). Its applicability to more subjective or creative tasks, such as detailed image captioning or visual storytelling, where the notion of a single "visually-grounded" reasoning chain is less clear, remains an open question.

**Methodological Assumptions and Hyperparameters.**     The dependency calculation at the core of VPPO is contingent on the choice of image perturbation method. Our ablation study (Appendix F) validates our choice of *Random Patch Blackening*, but it is plausible that the optimal masking strategy is task- or domain-dependent. Similarly, while our ablations (Subsection 5.2) identified optimal values for the key hyperparameters, i.e. the filtering ratio $k$ and the shaping range $[\beta_{\min}, \beta_{\max}]$, these values were determined on our specific training dataset and may require re-tuning when applying VPPO to new datasets or model scales to achieve maximum performance.

## L     ANALYSIS OF THE TRAINING DATASET

This section provides further details on the `ViRL39K` dataset (Wang et al., 2025a), which serves as the foundation for our reinforcement learning experiments. The choice of this dataset was deliberate, as its core properties align perfectly with the requirements for training a robust multimodal reasoning model.

**Topical Diversity and Reasoning Depth.**     A primary strength of `ViRL39K` is its broad topical diversity. The dataset is not confined to a single domain but instead contains approximately 39,000 queries spanning a wide range of challenging subjects, including mathematics, physics, chemistry, biology, and chart interpretation. This diversity is crucial for training a general-purpose reasoning model, as it prevents overfitting to a narrow task distribution and encourages the development of more fundamental, transferable reasoning skills.

**Suitability for Reinforcement Learning.**     The most critical feature of `ViRL39K` for our study is its verifiability. Every instance in the dataset is programmatically generated and comes with a definitive, unambiguous ground-truth answer. This property is indispensable for any RLVR framework, as it allows for the implementation of a clean, reliable, and automated reward function. By enabling a simple binary accuracy signal, it removes any need for subjective, model-based judges and ensures that the learning process is guided by objective correctness. For a comprehensive overview of the dataset's construction process and statistical breakdown, we refer the reader to the original publication.

## M     ANALYSIS OF EVALUATION BENCHMARKS

This section provides a brief analysis of the eight benchmarks used in our main evaluation. We deliberately selected this suite to cover a wide spectrum of challenges, from domain-specific mathematical skills to general logical cognition, ensuring a holistic assessment of our model's capabilities.

**Mathematical and Geometric Reasoning.**     This category forms the core of our evaluation, testing deep, domain-specific skills.

- **DynaMath** (Zou et al., 2024) is a unique benchmark designed to test the *robustness* of visual mathematical reasoning. Instead of using a static set of questions, it employs program-based generation to create numerous variants of seed problems, systematically altering numerical values and function graphs to challenge a model's ability to generalize rather than memorize.

- **Geo3k** (Lu et al., 2021) is a large-scale benchmark focused on high-school level *geometry*. Its key feature is the dense annotation of problems in a formal language, making it particularly well-suited for evaluating interpretable, symbolic reasoning approaches.

- **MathVerse** (Zhang et al., 2024) is specifically designed to answer the question: "Do MLLMs truly see the diagrams?" It tackles the problem of textual redundancy by providing six distinct versions of each problem, systematically shifting information from the text to the diagram. This allows for a fine-grained analysis of a model's reliance on visual versus textual cues.

- **MATH-Vision** (Wang et al., 2024a) elevates the difficulty by sourcing its problems from *real math competitions* (e.g., AMC, Math Kangaroo). Spanning 16 mathematical disciplines and 5 difficulty levels, it provides a challenging testbed for evaluating advanced, competition-level multimodal reasoning.

- **MMK12** (Meng et al., 2025) is a benchmark focused on K-12 level multimodal mathematical problems. It provides a strong test of foundational math reasoning skills that are essential for more advanced applications.

- **We-Math** (Qiao et al., 2024) introduces a novel, human-centric evaluation paradigm. It assesses reasoning by *decomposing composite problems into sub-problems* based on a hierarchy of 67 knowledge concepts. This allows for a fine-grained diagnosis of a model's specific strengths and weaknesses, distinguishing insufficient knowledge from failures in generalization.

**Logical Reasoning.** To assess more general cognitive abilities, we include a dedicated logical reasoning benchmark.

- **LogicVista** (Xiao et al., 2024) is designed to fill a critical gap by evaluating *general logical cognition* beyond the mathematical domain. It covers five core reasoning skills (inductive, deductive, numerical, spatial, and mechanical) across a variety of visual formats, testing the fundamental reasoning capabilities that underlie many complex tasks.

**Multi-discipline Reasoning.** Finally, to test performance on challenging, college-level problems that require true multimodal integration, we use a robust version of a well-known benchmark.

- **MMMU-Pro** (Yue et al., 2024) is a hardened version of the popular MMMU benchmark. It was specifically created to be unsolvable by text-only models by filtering out questions with textual shortcuts, augmenting the number of choices to reduce guessing, and introducing a vision-only format. It serves as a strong test of a model's ability to seamlessly integrate visual and textual information in a high-stakes, academic context.

## N  PROMPT TEMPLATE

For all training and evaluation experiments, we used the single, standardized prompt template shown below. Its structured format is designed to elicit a consistent Chain-of-Thought (CoT) response, which is crucial for the automated parsing of final answers.

---

**Reasoning Template**

**SYSTEM:**
You are a helpful assistant.

**USER:**
{question}

You first think through the reasoning process as an internal monologue, enclosed within <think> </think> tags. Then, provide your final answer enclosed within \boxed{}.

---

## O  QUALITATIVE CASE STUDIES: VPPO VS. BASELINE

To provide a more intuitive understanding of how VPPO improves reasoning performance, this section presents a qualitative analysis of three representative examples. For each case, we contrast the failure mode of the baseline with the correct reasoning process of our VPPO-7B model on the exact same problem. Notably, for each of these examples, our VPPO-7B model **produced the correct answer on all eight of its generation passes**, demonstrating the stability and robustness of its learned reasoning process.

These case studies are specifically chosen to highlight the practical impact of our hierarchical signal modulation. They illustrate how the baseline's uniform learning signal can lead to critical errors in visual perception or logical deduction, while VPPO's targeted approach successfully and consistently guides the model to the correct solution by focusing on pivotal, visually-grounded steps.

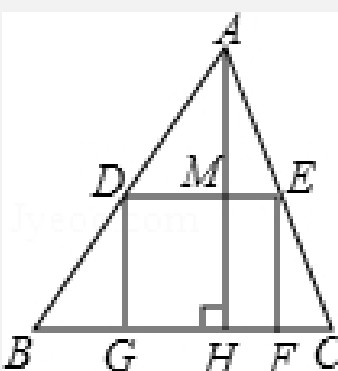

**Question**: As shown in the figure, a square DEFG model should be cut on a piece of triangle ABC paper. Among them, G and F are on BC, D and E are on AB and AC respectively, AH perpendicular BC and it intersects DE at M, if BC = 12.0, AH = 8.0, then the edge length of the square DEFG is ()

**Choices**: (A) $\frac{24}{5}$ cm   (B) 4cm   (C) $\frac{24}{7}$ cm   (D) 5cm

**Ground truth: (A)** $\frac{24}{5}$ **cm**

**<think>**

To solve for the edge length of the square DEFG, we need to leverage the properties of similar triangles and the given information.

1. **Identify the similarity of triangles:** Since DEFG is a square inscribed in the triangle ABC, $\triangle ADE$ and $\triangle ABC$ are similar by AA similarity (Angle-Angle similarity) criterion. This is because $\angle ADE = \angle ABC$ and $\angle DAE = \angle BAC$.

2. **Use the similarity ratio:** The ratio of the corresponding sides of similar triangles is equal. Therefore, we have:

$$\frac{DE}{BC} = \frac{AM}{AH}$$

3. **Express the given and unknown values:** Here, DE is the side length of the square we need to find, BC = 12 cm, and AH = 8 cm. We need to find the length of AM (the height from A to DE).

4. Since G and F are on BC, and E and D are on AB and AC respectively, we can use the fact that the height from A to DE is also the height from A to BC divided by the ratio of the side lengths of the triangles. The height from A to DE (AM) is the height from A to BC (AH) minus the height from M to BC.

5. Let's denote the side length of the square as $x$. Then, the height from A to DE (AM) is $8 - \frac{8x}{12} = 8 - \frac{2x}{3}$ (Incorrect expression for AM; it should be $AM = AH - MH = 8 - x$). Using the similarity ratio, we have:

$$\frac{x}{12} = \frac{8 - \frac{2x}{3}}{8}$$

6. **Solve the equation:**

$$\frac{x}{12} = 1 - \frac{x}{12}$$
$$\frac{x}{12} + \frac{x}{12} = 1$$
$$\frac{2x}{12} = 1$$
$$\frac{x}{6} = 1$$

$$x = \frac{24}{7} \text{ (Incorrect calculation; from the previous step } \frac{x}{6} = 1, \text{ the result should be } x = 6.)$$

Therefore, the edge length of the square DEFG is $\frac{24}{7}$ cm.

**</think>**

The correct answer is $\boxed{C}$.

### Generated Sample 1 by VPPO-7B

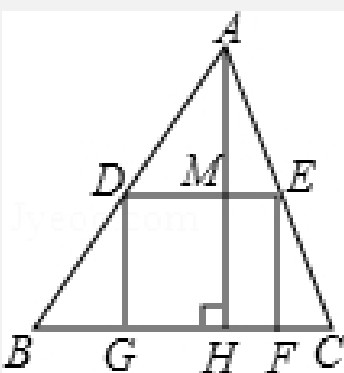

**Question**: As shown in the figure, a square DEFG model should be cut on a piece of triangle ABC paper. Among them, G and F are on BC, D and E are on AB and AC respectively, AH perpendicular BC and it intersects DE at M, if BC = 12.0, AH = 8.0, then the edge length of the square DEFG is ()

**Choices**: (A) $\frac{24}{5}$ cm  (B) 4cm  (C) $\frac{24}{7}$ cm  (D) 5cm

**Ground truth: (A) $\frac{24}{5}$ cm**

**<think>**

To determine the edge length of the square DEFG, we need to use the properties of similar triangles and the given dimensions of the triangle and the square. Here is the step-by-step reasoning:

1. **Identify the similar triangles:**

Since DEFG is a square, DE is parallel to BC, and therefore $\triangle ADE \sim \triangle ABC$ by the AA similarity criterion (Angle-Angle similarity). This gives us the ratio of the corresponding sides of the similar triangles.

2. **Set up the proportion:**

Since $\triangle ADE \sim \triangle ABC$, we have:

$$\frac{DE}{BC} = \frac{AM}{AH}$$

Let the side length of the square DEFG be $x$. Then, $DE = x$ and $AM = AH - MH = 8 - x$. Since $\frac{DE}{BC} = \frac{AM}{AH}$, we can write:

$$\frac{x}{12} = \frac{8-x}{8}$$

3. **Solve the equation:**

Cross-multiplying gives:

$$8x = 12(8 - x)$$

Expanding and simplifying:

$$8x = 96 - 12x$$
$$8x + 12x = 96$$
$$20x = 96$$
$$x = \frac{96}{20} = \frac{24}{5}$$

Therefore, the edge length of the square DEFG is $\frac{24}{5}$ cm.

**</think>**

The correct answer is $\boxed{A}$.

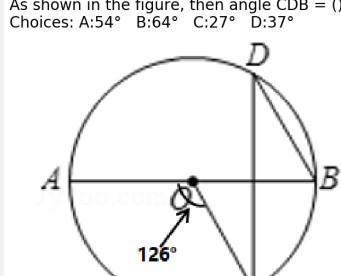

As shown in the figure, then angle CDB = ()
Choices: A:54°  B:64°  C:27°  D:37°

**Ground truth: (C) 27°**
**<think>**
To solve for the measure of angle $\angle CDB$, we need to use the properties of circles and the relationships between the angles.

1. Identify the type of angle given: The angle formed by the two radii at the center of the circle is $126°$.

2. Use the fact that the angle subtended by an arc at the center is twice the angle subtended by the same arc at any point on the circumference.

3. The angle at the center is $126°$, so the angle subtended by the same arc at point C (which is an inscribed angle) would be half of $126°$, which is $63°$. (Misapplication of the inscribed angle theorem; C is an endpoint of the arc, not a vertex viewing the arc.)

4. Since AB is a diameter, angle $\angle ACB$ is a right angle ($90°$).

5. Triangle $\triangle BCD$ is a right triangle with angle $\angle ACB$ as $90°$. (Incorrect statement; $\angle ACB$ is in $\triangle ABC$, and it is not an angle of $\triangle BCD$). The angle $\angle CDB$ is the angle we need to find.

The angle $\angle CDB$ is an inscribed angle that intercepts the same arc as the $63°$ angle at the center. Therefore, angle $\angle CDB$ is half of the supplementary angle of $126°$, which is $54°$. (Multiple logical errors: refers to a non-existent 63° central angle, incorrectly relates arcs, and the conclusion contradicts the premise.)

So, angle $\angle CDB = 54°$.
**</think>**
$\boxed{54}$.

**Generated Sample 2 by VPPO-7B**

As shown in the figure, then angle CDB = ()
Choices: A:54°  B:64°  C:27°  D:37°

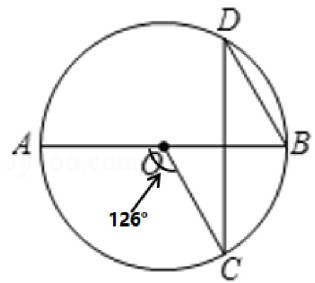

**Ground truth: (C) 27°**
**<think>**
To determine the measure of angle CDB, we need to use some properties of circles and the relationships between the angles in the circle. Here is the step-by-step reasoning:

1. **Identify the given information and the relevant theorems:**
- We are given that the measure of the central angle $\angle AOC$ is $126°$. According to the central angle theorem, the measure of the central angle is twice the measure of the inscribed angle that subtends the same arc.
- The measure of the inscribed angle $\angle ABC$ that subtends the same arc as the central angle $\angle AOC$ is half of the central angle. Therefore, $\angle ABC = \frac{1}{2} \times 126° = 63°$.

2. **Use the fact that the sum of the interior angles in a quadrilateral is $360°$:**
- Since AB is the diameter of the circle, $\angle ACB = 90°$ (angle in a semi-circle). Therefore, in $\triangle ABC$, we have:
$$\angle BAC = 180° - \angle ABC - \angle ACB = 180° - 63° - 90° = 27°.$$

3. **Relate the angles in the circle:**
- Since $\angle BAC$ and $\angle CDB$ are angles that subtend the same arc $BC$, they are equal. Therefore, $\angle CDB = \angle BAC = 27°$.

So, the measure of angle CDB is $\boxed{C}$.
**</think>**
The correct answer is $\boxed{C}$.

---

**Generated Sample 3 by VL-Rethinker-7B**

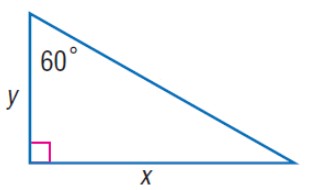

**Question**: In the triangle with hypotenuse length 20, find y.
**Choices**: (A) 4   (B) 6   (C) 8   (D) 10
**Ground truth: (D) 10**
**<think>**
To find the value of $y$ in the given right triangle, we can use the trigonometric relationship for the sine function (Incorrect function; cosine should be used as y is the adjacent side). The sine of an angle in a right triangle is defined as the length of the opposite side divided by the length of the hypotenuse.

Given:
- The hypotenuse $h = 20$
- The angle $\theta = 60°$
- The opposite side to the angle $\theta$ is $y$ (Incorrect identification; y is the side adjacent to the 60° angle, not opposite.)

The sine of $60°$ is given by:
$$\sin(60°) = \frac{y}{20}$$

(Incorrect formula based on the wrong side identification.)
We know that $\sin(60°) = \frac{\sqrt{3}}{2}$. Therefore:

$$\frac{\sqrt{3}}{2} = \frac{y}{20}$$

To solve for $y$, we can multiply both sides of the equation by 20:

$$y = 20 \times \frac{\sqrt{3}}{2}$$

$$y = 10\sqrt{3}$$

(This is the value for x, the opposite side, not y.)
Next, we simplify the expression. Using the approximate value of $\sqrt{3} \approx 1.732$:

$$y \approx 10 \times 1.732 = 17.32$$

However, we can now check the answer choices. The value $10\sqrt{3} \approx 17.32$ matches closest the provided options. (Illogical conclusion; 17.32 is not close to the integer options, and the chosen answer 'C' (8) does not follow from this value.)
**</think>**
$\boxed{C}$ .

**Generated Sample 3 by VPPO-7B**

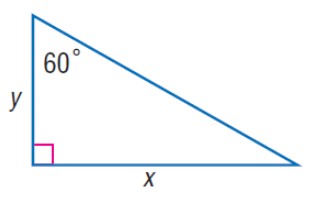

**Question**: In the triangle with hypotenuse length 20, find y.
**Choices**: (A) 4    (B) 6    (C) 8    (D) 10
**Ground truth: (D) 10**
**<think>**
To find the length of the side $y$ in the right triangle where the hypotenuse is 20 and the angle is 60 degrees, we can use the properties of a 30-60-90 triangle. In a 30-60-90 triangle, the sides are in the ratio $1 : \sqrt{3} : 2$, where the hypotenuse is the longest side and is twice the length of the side opposite the 30-degree angle.

Here, the hypotenuse is 20, which is twice the length of the side opposite the 30-degree angle. Therefore, the side opposite the 30-degree angle is half of the hypotenuse. Let's go through the steps:

1. Identify the given information: the hypotenuse is 20 and the angle is 60 degrees.
2. In a right triangle with a 60-degree angle, the side opposite the 30-degree angle is half the hypotenuse. Here, the side $y$ is the side opposite the 30-degree angle in the 30-60-90 triangle formed by the given angle.

The relationship in a 30-60-90 triangle is:

$$y = \frac{1}{2} \times \text{hypotenuse}$$

$$y = \frac{1}{2} \times 20$$
$$y = 10$$

Therefore, the length of $y$ is 10.
**</think>**
The correct answer is $\boxed{D}$.

