# OpenReview forum: "Spotlight on Token Perception for Multimodal Reinforcement Learning"
_ICLR.cc/2026/Conference — ICLR 2026 Poster_

### Official Review · Reviewer_4piz · 2025-10-29

**Soundness:** 4
**Presentation:** 3
**Contribution:** 3
**Rating:** 6
**Confidence:** 4

**Summary:**

This paper presents VPPO, a novel policy gradient algorithm for multimodal RL that improves LVLM reasoning by explicitly incorporating token-level visual perception. It identifies visually dependent tokens and trajectories to refine learning signals through advantage shaping and gradient filtering. VPPO demonstrates significant performance gains on diverse benchmarks, backed by theoretical analysis and extensive ablations.

**Strengths:**

1.The paper introduces a fresh and critical perspective by analyzing and leveraging token-level visual perception in multimodal RL, addressing a clear gap in existing "modality-agnostic" methods. The insights into sparse token dependency and heterogeneous trajectory grounding are well-motivated.
2. VPPO consistently achieves new state-of-the-art results on a comprehensive suite of challenging multimodal reasoning benchmarks, showing substantial accuracy gains over leading baselines across different model scales (7B and 32B). This robust performance is a key highlight.

**Weaknesses:**

1.the additional computational cost of a second forward pass for token perception calculation needs more detailed quantification across different model scales (e.g., specific training times or throughput comparisons for 32B models).
2. Evaluation is restricted to reasoning-intensive benchmarks. The method's generalizability to other multimodal tasks, larger model scales, and different architectures remains unexplored, particularly given its reliance on a specific image perturbation strategy.

**Questions:**

Please refer to the Weaknesses section above for details.

---

> ### Author Response · Authors · 2025-11-20
> **Response to Reviewer 4piz (Part 1)**
>
> We sincerely thank the reviewer for highlighting our novel perspective on token-level visual dependency in multimodal RL. In our response, we will address each of the weaknesses (W1-W2) in detail.
>
> ---
>
> > W1: the additional computational cost of a second forward pass for token perception calculation needs more detailed quantification across different model scales.
>
> Thanks for your concern. We have conducted a detailed empirical analysis to quantify this overhead and provide a direct performance comparison under an equal computational budget.
>
> First, to precisely measure the overhead introduced by VPPO's second forward pass, we compared the total training time and throughput against the DAPO baseline for both our 7B and 32B models.
>
> The results are presented below:
>
> | Model Scale | Method | Total Training Time (hours) | Training Throughput (samples/sec) | Overhead (%) |
> | :--- | :--- | :---: | :---: | :---: |
> | **7B** | DAPO (Baseline) | 15.5 | ~1.39 | - |
> | (8x H800) | **VPPO (Ours)** | **17.0** | **~1.27** | **+9.7%** |
> | | | | | |
> | **32B** | DAPO (Baseline) | 91.2 | ~0.24 | - |
> | (32x H800) | **VPPO (Ours)** | **100.3** | **~0.22** | **+10.0%** |
>
> Benefiting from the fact that the probabilities of all tokens within a known response can be calculated in a single, parallel process, our measurements confirm that the overhead is a modest and consistent ~10% across both model scales, making it a predictable and manageable increase.
>
> While the overhead is minor, we recognize that a more rigorous evaluation involves comparing performance under a fixed time budget. To this end, we trained the 7B DAPO baseline for an extended 17.0 hours, matching the exact training time of our VPPO-7B model. This equal-budget comparison ensures that any performance difference is attributable to the method's efficiency, not additional training time.
>
> The results are presented below:
>
> | Method (7B Model) | Time | MathVerse | DynaMath | MMK12 | Geo3k | MathVision | We-Math | LogicVista | MMMU-Pro | **Avg.** |
> | :--- | :---: | :---: | :---: | :---: | :---: | :---: | :---: | :---: | :---: | :---: |
> | DAPO (Baseline) | 15.5h | 68.3 | 66.6 | 82.1 | 41.5 | 30.5 | 68.0 | 46.8 | 35.9 | 55.0 |
> | DAPO (Equal-Budget) | **17.0h** | 68.6 | 67.0 | 81.9 | 42.1 | 30.6 | 67.6 | 46.2 | 36.3 | 55.0 |
> | **VPPO (Ours)** | **17.0h** | **71.6** | **68.1** | **82.8** | **46.5** | **33.3** | **71.5** | **47.9** | **37.9** | **57.5** |
>
> Theese results demonstrate that VPPO's performance gains are not an artifact of longer training but stem from a fundamental improvement in **learning efficiency**. By shaping the learning signal at both the trajectory and token levels, VPPO enables the model to acquire complex reasoning skills more effectively within the same time budget. These findings validate that the minor computational cost is a highly effective trade-off for the substantial and broad-based improvements in multimodal reasoning.

---

> > ### Author Response · Authors · 2025-11-20
> > **Response to Reviewer 4piz (Part 2)**
> >
> > > W2: The method's generalizability, particularly given its reliance on a specific image perturbation strategy.
> >
> > Thanks for your comment. We have conducted new experiments that address your concerns, demonstrating that VPPO is both scalable and generalizes well to broader tasks.
> >
> > * Generalization to Broader Multimodal Tasks
> >
> > To test VPPO's effectiveness beyond reasoning, we evaluated our 7B model on two unseen, general-purpose VQA benchmarks. The results show that not only does VPPO **not** degrade general VQA performance, but it actually achieves the highest scores among all RL-tuned methods. This suggests that by focusing on perceptually pivotal tokens, VPPO builds a more robust visual grounding that transfers positively to other vision-language tasks.
> >
> > The results are presented below:
> >
> > | Model                  | A-OKVQA-val (%) | SimpleVQA-EN (%) | Average (%) |
> > | :--------------------- | :-------------: | :--------------: | :---------: |
> > | Qwen2.5-VL-7B (Base)   | 84.2            | 38.6             | 61.4        |
> > | + GRPO                 | 87.4            | 43.1             | 65.3        |
> > | + DAPO                 | 87.9            | 42.9             | 65.4        |
> > | **+ VPPO (Ours)**      | **87.9**        | **43.8**         | **65.9**    |
> >
> > * Scalability to Larger Models
> >
> > To confirm that VPPO's benefits are not limited to a specific model size, we extended our evaluation to the Qwen2.5-VL-32B model. The results below show that VPPO consistently outperforms the baselines, achieving the highest average accuracy and demonstrating its effective scalability.
> >
> > The results are presented below:
> >
> > | Model | MathVerse | DynaMath | MMK12 | Geo3k | MathVision | We-Math | LogicVista | MMMU-Pro | **Avg.** |
> > | :--- | :---: | :---: | :---: | :---: | :---: | :---: | :---: | :---: | :---: |
> > | Qwen2.5-VL-32B | 68.5 | 68.7 | 68.8 | 47.0 | 39.3 | 71.0 | 52.8 | 39.6 | 57.0 |
> > | + GRPO | 74.2 | 71.6 | 80.7 | 51.4 | 42.8 | 76.7 | 58.3 | 45.4 | 62.6 |
> > | + DAPO | 73.3	| 72.6 | 86.4 | 51.4 | 42.8 | 76.2 | 58.9 | 46.4 | 63.5 |
> > | **+ VPPO (Ours)** | **75.1** | **73.1** | 86.3 | **53.4** | **44.6** | **77.7** | **59.2** | **47.1** | **64.6** |
> >
> > These new results provide strong evidence for VPPO's generalization and scalability, and our ablation studies validate its core methodological choices.
> >
> > ---
> >
> > We have incorporated all new experiments in the revised manuscript. Specifically, the analysis of computational overhead can be found in the Appendix J, the generalization to out-of-domain VQA can be found in Section 5.3, and the comprehensive 32B results can be found in the main Table 1. Thank you again for your time and insightful feedback.

---

### Official Review · Reviewer_CS1Y · 2025-11-01

**Soundness:** 3
**Presentation:** 3
**Contribution:** 3
**Rating:** 6
**Confidence:** 2

**Summary:**

This paper introduces Visually-Perceptive Policy Optimization (VPPO), a novel reinforcement learning algorithm for multimodal reasoning that addresses the inefficiency of uniform learning signals in existing approaches. The authors identify that current multimodal RLVR methods fail to distinguish between tokens with different visual dependencies, leading to suboptimal learning. VPPO introduces two key mechanisms: (1) Token-level Gradient Filtering (TGF) that focuses updates on visually-dependent tokens, and (2) Trajectory-level Advantage Shaping (TAS) that reweights advantages based on overall visual dependency. Evaluated on eight benchmarks using Qwen2.5-VL models, VPPO achieves 19.2% and 7.6% average accuracy improvements for 7B and 32B models respectively.

**Strengths:**

- Thorough analysis: The paper provides excellent insights into token-level visual dependency distributions and their impact on RL learning.

- Strong empirical results: Consistent improvements across 8 benchmarks and 2 model scales demonstrate robustness.

- Faster convergence: Figure 5 shows VPPO not only achieves better final performance but converges faster

**Weaknesses:**

I see no major weakness. Thanks for the authors' hard work.

**Questions:**

I am not an expert in this area, and I am willing to discuss with other reviewers.
- After applying the proposed method, will the model diminish its capability on general visual language tasks, such as VQA?

---

> ### Author Response · Authors · 2025-11-20
> **Response to Reviewer CS1Y**
>
> We sincerely thank the reviewer for recognizing our novel algorithm improves both performance and training efficiency. In our response, we will address the question in detail.
>
> ---
>
> > **Q: After applying the proposed method, will the model diminish its capability on general visual language tasks, such as VQA?**
>
> We thank the reviewer for this question. To provide a data-driven answer, we conducted a new set of experiments on two unseen, out-of-domain Visual Question Answering (VQA) benchmarks: A-OKVQA-val (\~1.1k items) and SimpleVQA-EN (\~1k items). We evaluated the Qwen2.5-VL-7B base model against versions fine-tuned with GRPO, DAPO, and our VPPO, using the same evaluation protocol as in our main experiments (average accuracy over 8 generation passes).
>
> * The results are summarized in the table below:
>
> | Model                  | A-OKVQA-val (%) | SimpleVQA-EN (%) | Average (%) |
> | :--------------------- | :-------------: | :--------------: | :---------: |
> | Qwen2.5-VL-7B (Base)   | 84.2            | 38.6             | 61.4        |
> | + GRPO                 | 87.4            | 43.1             | 65.3        |
> | + DAPO                 | 87.9            | 42.9             | 65.4        |
> | **+ VPPO (Ours)**      | **87.9**        | **43.8**         | **65.9**    |
>
> These results offer two clear insights. First, far from diminishing general capabilities, all RL fine-tuning methods improve performance over the base model by a significant margin (approx. +4% on average). This demonstrates a positive transfer of skills learned during complex reasoning to general visual understanding tasks.
>
> Second, VPPO achieves the highest overall performance. We attribute this superior generalization to its core mechanism. By focusing learning on *perceptually pivotal tokens*, VPPO strengthens the model’s fundamental visual grounding, an improvement that robustly benefits general-purpose VQA tasks.
>
> These experiments confirm that our method not only excels at its primary reasoning tasks but also generalizes more effectively than alternatives to out-of-domain benchmarks.
>
> ---
> We have incorporated the new experiment into Section 5.3 in the revised manuscript. Thank you again for your time and insightful feedback.

---

### Official Review · Reviewer_iQRs · 2025-11-02

**Soundness:** 4
**Presentation:** 3
**Contribution:** 3
**Rating:** 6
**Confidence:** 3

**Summary:**

This paper introduces VPPO, a variant of GRPO that encourages rollouts with higher visual dependence. It adopts a framework similar to PAPO, contributing several additional insights. It gives a proper measurement of visual dependence for each rollout and reveals two valuable insights with empirical evidence. Building upon these, it proposes two tailored adaptations to GRPO that cover both micro and macro hierarchy. Finally, comprehensive and detailed experiments validate the proposed modules, forming the paper as a coherent and insightful work.

**Strengths:**

1. **Well-presented Paper** The paper conducts a step-by-step and coherent exploration, from intuitive motivation, empirical evidence, tailored solutions, and comprehensive experiments. Each part is presented clearly and soundly.
2. **Insightful Empirical Findings**. The paper introduces a measurement for visual dependence. Building on this metric, the paper demonstrates that only a small proportion of tokens are highly visually dependent, and the overall visual dependency is varied across rollouts. This metric and findings might inspire further research.
3. **Theoretical Guarantee**. The paper proves that VPPO constructs a lower-variance policy gradient estimator, guaranteeing overall training stability.
4. **Comprehensive Experiments**. There are very comprehensive experiments studying various aspects of the proposed methods. They collectively validate the effectiveness and robustness of VPPO.

**Weaknesses:**

1. **Solid yet Expected Contributions**. Despite the overall high quality, VPPO inherits the framework from PAPO and makes incremental improvements. This leads to the paper's contributions being sufficient but not groundbreaking.
2. **Minor Gaps between Insights and Solutions**. For the first insight, the paper demonstrates that highly visually dependent tokens are sparsely distributed across rollouts. VPPO then computes gradients exclusively yet evenly on top-ranked tokens. The paper does not discuss the reason behind adopting the discrete binary mask instead of a continuous soft mask based on the dependent value. For the second insight, the paper shows that the overall visual dependency is varied across rollouts. However, it lacks an empirical link between visual dependency and reasoning quality. Additionally, the amplification term is a bit confusing: in cases where a rollout exhibits a high visual dependency but yields incorrect results, the term would amplify discouragement, which somewhat misaligns with motivation.

**Questions:**

1. As in weakness 2, why do you adopt a discrete binary mask operation instead of a continuous soft mask based on the dependent value?
2. What are the intuitive benefits of the amplification term on the advantages? Why do you choose not to reflect the overall visual dependency on the reward term?
3. Can you further clarify the formulation difference between VPPO and PAPO?

---

> ### Author Response · Authors · 2025-11-20
> **Response to Reviewer iQRs (Part 1)**
>
> We sincerely thank the reviewer for recognizing our novel visual dependency metric and the insightful empirical findings. We address each of the weaknesses (W1-W2) and questions (Q1-Q3) in detail below.
>
> ---
>
> > **W1: Solid yet Expected Contributions. Despite the overall high quality, VPPO inherits the framework from PAPO and makes incremental improvements. This leads to the paper's contributions being sufficient but not groundbreaking.**
>
> We thank the reviewer for this question. We would like to further clarify the nature of our contribution, which tackles a core bottleneck in multimodal reinforcement learning.
>
> The central challenge in applying RL to multimodal reasoning is the **misalignment between the learning signal and the task's inherent structure**. Standard RLVR frameworks broadcast a uniform, trajectory-level reward to every token in a generated response. Previous multimodal reasoning works, including PAPO, ignores that visual reasoning is not a uniform process.
>
> Instead, our VPPO makes the **first attempt to restruct the learning signal** to align with the perceptual demands of the visual task, which is fundamentally different from the principle of previous works. Specifically, VPPO intervenes directly within the policy gradient calculation. It internally re-weights learning based on a trajectory's visual grounding and, more critically, concentrates the gradient updates exclusively on the pivotal tokens where vision and language intersect.
>
> ---
>
> > **W2: (1) The paper does not discuss the reason behind adopting the discrete binary mask instead of a continuous soft mask based on the dependent value.**
>
> We thank the reviewer for this question. To empirically validate our choice, we conducted a new ablation study directly comparing our discrete binary mask against a continuous soft mask.
>
> For this experiment, the soft mask assigns an adaptive weight $w_t$ to each token's gradient based on its normalized visual dependency score. To ensure a fair comparison, the method is carefully calibrated so that the *average* weight across all tokens in a trajectory matches our target filtering ratio (k=0.4). The process involves three steps:
> 1.  **Z-Score Normalization:** We first normalize the raw dependency scores $S_t$ within each trajectory into Z-scores: $Z_t = (S_t - \mu_S) / (\sigma_S + \epsilon)$.
> 2.  **Offset Calibration:** We then find a unique offset $c$ that satisfies the constraint $\frac{1}{N} \sum_{t=1}^{N} \text{sigmoid}(Z_t - c) = \mu_{\text{target}}$.
> 3.  **Weight Generation:** The final weight for each token is calculated as $w_t = \text{sigmoid}(Z_t - c)$.
>
> * Performance Comparison of Binary Mask vs. Soft Mask on 7B Models
>
> | Model Configuration | MathVerse | DynaMath | MMK12 | Geo3k | MathVision | We-Math | LogicVista | MMMU-Pro | **Avg.** |
> |:---|:---:|:---:|:---:|:---:|:---:|:---:|:---:|:---:|:---:|
> | Baseline (DAPO) | 68.3 | 66.6 | 82.1 | 41.5 | 30.5 | 68.0 | 46.8 | 35.9 | 55.0 |
> | VPPO w/ Soft Mask | 70.0 | 67.2 | 82.6 | 43.8 | 32.6 | 70.6 | 46.6 | 36.3 | 56.2 |
> | **VPPO (Binary Mask)** | **71.6** | **68.1** | **82.8** | **46.5** | **33.3** | **71.5** | **47.9** | **37.9** | **57.5** |
>
> The results clearly show that while the soft mask outperforms the baseline by 1.2% on average, the **discrete binary mask consistently achieves the best performance, surpassing the soft mask by 1.3% and the baseline by a total of 2.5% on average.**
>
> We hypothesize this is because the binary mask acts as a more decisive noise filter. By completely removing gradients from non-pivotal tokens, it provides a stronger, more focused learning signal, forcing the model to concentrate its updates on the most critical moments of visually-grounded reasoning. Therefore, our approach is more potent regularizer than the graded signal from the soft mask.

---

> > ### Author Response · Authors · 2025-11-20
> > **Response to Reviewer iQRs (Part 2)**
> >
> > > **W2: (2) However, it lacks an empirical link between visual dependency and reasoning quality. Additionally, the amplification term is a bit confusing: in cases where a rollout exhibits a high visual dependency but yields incorrect results, the term would amplify discouragement, which somewhat misaligns with motivation.**
> >
> > We thank the reviewer for this question. The motivation of our Trajectory-level Advantage Shaping (TAS) is not that high visual dependency directly correlates with correctness, but that it indicates a **perceptually informative training sample**, from which the model can effectively learn.
> >
> > *   A **high-dependency, correct** trajectory is a gold-standard example of successful visual reasoning. Amplifying its positive advantage strongly reinforces a desirable, visually-grounded strategy.
> >
> > *   To address the reviewer's specific concern: a **high-dependency, incorrect** trajectory represents a critical failure *within* a visual reasoning process, not a language-based guess. This is highly informative because it pinpoints a specific flaw in the model's perception or logic. By amplifying its penalty, we provide a powerful and targeted signal to correct that exact mistake.
> >
> > In essence, TAS works by amplifying the learning signal from the most perceptually informative samples (both positive and negative) while dampening the signal from low-dependency trajectories. This prevents the model from learning non-robust shortcuts and focuses it on improving its genuine visual reasoning abilities.
> >
> > ---
> >
> > > **Q1: As in weakness 2, why do you adopt a discrete binary mask operation instead of a continuous soft mask based on the dependent value?**
> >
> > We thank the reviewer for this question. As detailed in our response to W2(1), we conducted a new ablation study to address this question. The results empirically confirmed that the discrete binary mask provides a stronger learning signal and yields superior performance compared to a continuous soft mask. We have included the full details and result table in the W2(1) section above.
> >
> > ---
> >
> > > **Q2: What are the intuitive benefits of the amplification term on the advantages? Why do you choose not to reflect the overall visual dependency on the reward term?**
> >
> > We thank the reviewer for this question. As discussed in W2(2), the benefit of amplifying the advantage is to prioritize learning from the most informative, visually-grounded trajectories.
> >
> > Additionally, we modulate the **advantage** rather than the **reward** because it provides a more stable and direct mechanism for guiding the policy update. To validate this, we ran a new experiment directly comparing our method against applying the same modulation to the reward term ("Reward Shaping").
> >
> > * Performance Comparison of Advantage Shaping vs. Reward Shaping on 7B Models:
> >
> > | Model Configuration | MathVerse | DynaMath | MMK12 | Geo3k | MathVision | We-Math | LogicVista | MMMU-Pro | **Avg.** |
> > |:---|:---:|:---:|:---:|:---:|:---:|:---:|:---:|:---:|:---:|
> > | DAPO (Baseline) | 68.3 | 66.6 | 82.1 | 41.5 | 30.5 | 68.0 | 46.8 | 35.9 | 55.0 |
> > | VPPO w/ Reward Shaping | 70.7 | 68.4 | 82.6 | 44.7 | 33.5 | 70.1 | 47.1 | 37.6 | 56.8 |
> > | **VPPO w/ Advantage Shaping** | **71.6** | 68.1 | **82.8** | **46.5** | 33.3 | **71.5** | **47.9** | **37.9** | **57.5** |
> >
> > The results confirm that shaping the advantage is more effective, outperforming the reward shaping strategy by **0.7%** on average. We attribute this to the **stability of the policy gradient update**: modifying the reward introduces our heuristic *before* the advantage calculation, which can lead to higher variance and a noisier learning signal. In contrast, by directly modulating the final **advantage** term, we apply a clean, deterministic scaling to the pre-calculated gradient. This allows us to more reliably control the update magnitude, resulting in a more stable and effective learning process.

---

> ### Author Response · Authors · 2025-11-20
> **Response to Reviewer iQRs (Part 3)**
>
> > **Q3: Can you further clarify the formulation difference between VPPO and PAPO?**
>
> We thank the reviewer for this question. Here we pinpoint the core mathematical and structural distinctions between our work and PAPO. The difference is not incremental, but rather a shift from an **additive auxiliary loss** to **direct signal modulation**.
>
> First, let's examine the objective functions. PAPO's objective introduces a separate, external perception loss that competes with the primary RL objective:
>
> $$
> J_{\text{PAPO}}(\theta) = J_{\text{GRPO}}(\theta) + \underbrace{\gamma D\_{KL}[\pi\_\theta || \pi\_\theta^{\text{mask}}]}_{\text{Auxiliary Perception Loss}} - \text{Regularizers}
> $$
>
> As shown, the learning signal in PAPO is a sum of two distinct gradients: one from the task-oriented GRPO objective and another from the auxiliary KL divergence term. This term operates at the coarse **trajectory level**, encouraging the entire policy to diverge from one that ignores the visual input.
>
> In contrast, VPPO’s formulation does not add any new loss term. Instead, it intervenes *within* the core policy gradient calculation to directly reshape the learning signal itself:
>
> $$
> \mathcal{L}_{\text{VPPO}}(\theta) = \mathbb{E} \left[ \sum\_{t} \underbrace{m\_{i,t}}\_{\text{TGF}} \cdot \log\pi\_\theta(o\_{i,t}|s\_{i,t}) \cdot (\underbrace{\alpha(\tau\_i)}\_{\text{TAS}} \cdot \hat{A}\_{\text{GRPO}, i}) \right]
> $$
>
> Here, the modulation is internal and precise. The trajectory's advantage ($\hat{A}\_{\text{GRPO}, i}$) is directly re-weighted by its overall visual dependency ($\alpha(\tau_i)$), and the binary mask ($m_{i,t}$) ensures that gradients are only computed for the few perceptually pivotal tokens.
>
> This mathematical distinction is crucial. While PAPO treats perception as an external, competing objective, VPPO integrates it as a direct, **hierarchical modulator** of the primary learning signal.
>
> This is achieved on two levels. At the **trajectory level**, our **Advantage Shaping (TAS)** re-weights each path's advantage ($\alpha(\tau_i)$) to prioritize learning from visually-grounded reasoning. Subsequently, at the **token level**, our **Gradient Filtering (TGF)** solves the critical signal dilution problem by concentrating this shaped learning signal ($m_{i,t}$) exclusively on the pivotal moments where vision grounds the reasoning process.
>
> This mechanism of directly shaping and focusing the core learning signal is fundamentally different and more targeted than adding an auxiliary loss.
>
> ---
>
> We have incorporated all new experiments in the revised manuscript. Specifically, the ablation study of binary mask can be found in Section 5.2 and Table 6,  and the ablation study of advantage shaping can be found in Section 5.2 and Table 7. Thank you again for your time and insightful feedback.

---

> > ### Comment · Reviewer_iQRs · 2025-11-26
> >
> > Thank you very much for the detailed response! My concerns regarding methodology are addressed with the additional results provided. Therefore, I have raised my score to 8.

---

> > > ### Author Response · Authors · 2025-11-26
> > > **Response to Reviewer iQRs**
> > >
> > > Thank you for your positive feedback. We sincerely appreciate your time and recognition.

---

### Official Review · Reviewer_paED · 2025-11-05

**Soundness:** 3
**Presentation:** 4
**Contribution:** 3
**Rating:** 6
**Confidence:** 3

**Summary:**

The paper explores how tokens generated during visual reasoning influence the post-training of large language models. The core motivation is to understand which reasoning tokens are most affected when noise is introduced into the image. Through empirical analysis, the authors find that a small subset of tokens plays a disproportionately important role in the reasoning process, and accurately predicting these tokens is essential. Building on this insight, they propose a method for visual reasoning that is validated through experiments and ablation studies.

**Strengths:**

1. The paper is well written and formatted nicely; it is easy to understand the core message of the paper.
2. The authors present a well-informed set of experiments and ablations that help show the effectiveness of their method.
3. The paper's motivation is well rooted in recent literature evaluating the effectiveness of reasoning traces during post-training of LLMs [1,2].

[1] Beyond the 80/20 Rule: High-Entropy Minority Tokens Drive Effective Reinforcement Learning for LLM Reasoning

[2] TreeRL: LLM Reinforcement Learning with On-Policy Tree Search

**Weaknesses:**

1. A key concern is the sensitivity of the proposed method to the Entropy Loss parameter introduced for DAPO. This parameter requires careful tuning to achieve optimal performance.

2. While the authors cite [1] and acknowledge related work in LLM reasoning, the claim that “in the multimodal domain, a pivotal token is not just a logical fork but a critical moment of visually grounded reasoning” lacks sufficient support, as no direct comparison is made with those prior methods.

3. It would have been valuable to include results comparing VPPO, GRPO, and DAPO on the 32B model, as this could provide deeper insight into their relative effectiveness at scale.

**Questions:**

1. The sensitivity of the method to the entropy loss hyperparameter appears to stem from VPPO being built on top of DAPO. A crucial experiment would be to substitute DAPO with GRPO within VPPO to observe how the results change. This would help disentangle the specific contributions of VPPO from those of the underlying policy optimization algorithm.

2. Additionally, following [1], are the tokens with the highest KL divergence also the ones with the highest entropy? Clarifying this connection would help align the findings with recent literature and enhance the interpretability of the pivotal token analysis.

---

> ### Author Response · Authors · 2025-11-20
> **Response to Reviewer paED (Part 1)**
>
> We sincerely thank the reviewer for recognizing our well-informed set of experiments and ablations. We have carefully considered all points and provide detailed responses to the weaknesses (W1-W3) and questions (Q1-Q2) below.
>
> ---
>
> > **W1: A key concern is the sensitivity of the proposed method to the Entropy Loss parameter introduced for DAPO. This parameter requires careful tuning to achieve optimal performance.**
>
> We thank the reviewer for this question. The entropy penalty was introduced primarily to stabilize the DAPO baseline, ensuring a fair and robust comparison. As detailed in our ablation study in Appendix E (Figure 9), the model's performance stabilizes and shows low sensitivity to the specific coefficient value once it reaches 0.04 or greater. This indicates that while the penalty is crucial for preventing policy collapse, the model is not sensitive to the precise value chosen within this effective range.
>
> Crucially, to ensure a fair comparison, we used a fixed coefficient (0.06) for both the DAPO baseline and our VPPO method across all comparative experiments. This controlled setup guarantees that the performance gains we report are attributable to VPPO's innovations, rather than variations in this stabilization parameter.
>
> ---
>
> > **W2: While the authors cite [1] and acknowledge related work in LLM reasoning, no direct comparison is made with those prior methods.**
>
> We appreciate the reviewer's suggestion to clarify this comparison. We have performed a direct comparison with entropy-based method [1] in Section 5.2 and Table 4. For ease of reference, we have reproduced the results from Table 4 below.
>
> | Filtering Mechanism | MathVerse | DynaMath | MMK12 | Geo3k | MathVision | We-Math | LogicVista | MMMU-Pro | Avg. |
> | :--- | :---: | :---: | :---: | :---: | :---: | :---: | :---: | :---: | :---: |
> | Baseline (DAPO) | 68.3 | 66.6 | 82.1 | 41.5 | 30.5 | 68.0 | 46.8 | 35.9 | 55.0 |
> | + Random (k = 0.4) | 69.3 | 66.2 | 76.8 | 42.0 | 31.0 | 69.3 | 47.5 | 36.2 | 54.8 |
> | + Entropy (k = 0.2) | 70.1 | 67.2 | 77.9 | 45.0 | 32.6 | 70.6 | 48.0 | 36.4 | 56.0 |
> | + Entropy (k = 0.4) | 69.3 | 67.6 | 80.0 | 42.8 | 31.7 | 69.4 | 47.4 | 37.0 | 55.7 |
> | + Entropy (k = 0.6) | 69.9 | 67.4 | 81.0 | 43.4 | 31.4 | 69.1 | 47.1 | 36.9 | 55.8 |
> | + Entropy (k = 0.8) | 69.6 | 66.9 | 81.1 | 41.6 | 31.2 | 69.0 | 46.6 | 36.2 | 55.3 |
> | **+ Visual Dependency (k = 0.4)** | **71.2** | **68.6** | 80.9 | **45.3** | **34.7** | 70.3 | **48.2** | **37.3** | **57.1** |
>
> The results clearly show that filtering gradients using Visual Dependency is more effective than using predictive entropy. On average, Visual Dependency achieves a score of **57.1**, which is **1.1** points higher than the best entropy-based configuration (56.0) and **2.1** points above the DAPO baseline. This quantitative evidence strongly supports our claim: in multimodal reasoning, visual dependency is a more precise and effective signal than predictive entropy for identifying the critical tokens that should guide policy updates.
>
> [1] Beyond the 80/20 Rule: High-Entropy Minority Tokens Drive Effective Reinforcement Learning for LLM Reasoning
>
> ---
>
> > **W3: It would have been valuable to include results comparing VPPO, GRPO, and DAPO on the 32B model, as this could provide deeper insight into their relative effectiveness at scale.**
>
> We thank the reviewer for this suggestion. In response, we have conducted the requested experiments comparing VPPO, GRPO, and DAPO on the 32B model. The results, presented in the table below, confirm that VPPO maintains its significant performance advantage at a larger scale, underscoring its scalability.
>
> * Performance comparison of 32B models (avg@8 acc %):
>
> | Model | MathVerse | DynaMath | MMK12 | Geo3k | MathVision | We-Math | LogicVista | MMMU-Pro | **Avg.** |
> | :--- | :---: | :---: | :---: | :---: | :---: | :---: | :---: | :---: | :---: |
> | Qwen2.5-VL-32B | 68.5 | 68.7 | 68.8 | 47.0 | 39.3 | 71.0 | 52.8 | 39.6 | 57.0 |
> | + GRPO | 74.2 | 71.6 | 80.7 | 51.4 | 42.8 | 76.7 | 58.3 | 45.4 | 62.6 |
> | + DAPO | 73.3	| 72.6 | 86.4 | 51.4 | 42.8 | 76.2 | 58.9 | 46.4 | 63.5 |
> | **+ VPPO (Ours)** | **75.1** | **73.1** | 86.3 | **53.4** | **44.6** | **77.7** | **59.2** | **47.1** | **64.6** |
>
> The results confirm VPPO's superior performance at scale. VPPO achieves the highest average score of **64.6%**, surpassing both DAPO (**+1.1** points) and GRPO (**+2.0** points). These findings provide strong evidence that VPPO is an effective and scalable method for enhancing multimodal reasoning.

---

> > ### Author Response · Authors · 2025-11-20
> > **Response to Reviewer paED (Part 2)**
> >
> > > **Q1: A crucial experiment would be to substitute DAPO with GRPO within VPPO to observe how the results change. This would help disentangle the specific contributions of VPPO from those of the underlying policy optimization algorithm.**
> >
> > We thank the reviewer for this question. We have conducted a new set of experiments implementing VPPO on top of GRPO. The results in the table below show a consistent trend: `VPPO w/ GRPO` improves upon `GRPO` by **1.7%**, and `VPPO w/ DAPO` improves upon `DAPO` by **2.5%**. This demonstrates that VPPO functions as a general and effective enhancement, independent of the underlying policy optimization algorithm.
> >
> > * Performance comparison of 7B models (avg@8 acc %):
> >
> > | Model Configuration | MathVerse | DynaMath | MMK12 | Geo3k | MathVision | We-Math | LogicVista | MMMU-Pro | **Avg.** |
> > | :--- | :---: | :---: | :---: | :---: | :---: | :---: | :---: | :---: | :---: |
> > | Qwen2.5-VL-7B | 39.0 | 55.7 | 42.5 | 37.1 | 18.4 | 46.4 | 42.4 | 25.1 | 38.3 |
> > | + GRPO | 66.5 | 65.8 | 72.3 | 40.2 | 30.7 | 68.1 | 45.6 | 35.2 | 53.1 |
> > | **+ VPPO w/ GRPO** | **69.7** | **66.4** | **76.4** | **41.0** | **31.7** | **69.5** | **47.6** | **35.8** | **54.8** |
> > | + DAPO | 68.3 | 66.6 | 82.1 | 41.5 | 30.5 | 68.0 | 46.8 | 35.9 | 55.0 |
> > | **+ VPPO w/ DAPO** | **71.6** | **68.1** | **82.8** | **46.5** | **33.3** | **71.5** | **47.9** | **37.9** | **57.5** |
> >
> > ---
> >
> > > **Q2: Are the tokens with the highest KL divergence also the ones with the highest entropy? Clarifying this connection would help align the findings with recent literature and enhance the interpretability of the pivotal token analysis.**
> >
> > We thank the reviewer for this question. Our analysis shows a significant overlap (~80%) between tokens with high visual dependency and those with high entropy. However, the key advantage of our method lies in the crucial tokens that visual dependency uniquely identifies.
> >
> > *   **Critical Reasoning Steps (High Dependency & High Entropy):**
> >     These are tokens at critical reasoning junctions where the model is uncertain and relies on visual information. Examples include mathematical operators like **`=`**, words applying geometric theorems like **`twice`** or **`parallel`**, and logical connectors such as **`Since`** or **`Thus`**. These tokens initiate or execute a reasoning step based on visual evidence, a point where the model may be uncertain. Both our method and entropy-based methods correctly identify them as important.
> >
> > *   **Visually-Grounded Facts (High Dependency & Low Entropy):**
> >     This is where our method excels. These tokens represent direct, factual observations from the image that the model is confident about once perceived (hence, low entropy). Examples include specific numbers like **`25°`** or **`126`**, or names of geometric shapes like **`△AOB`**. These tokens are critical because they serve as the factual premises for the entire reasoning process. Without correctly perceiving these foundational visual facts, the reasoning chain cannot begin correctly. Entropy-based methods often overlook these tokens, but our visual dependency metric rightly identifies their importance.
> >
> > In summary, as shown in literature such as [1], high-entropy tokens mark points of logical uncertainty, which is effective for text-based reasoning. However, for multimodal tasks, it is also important to reward the model for correctly perceiving the foundational visual facts. By capturing both types of pivotal tokens, VPPO builds its reasoning on a more solid perceptual foundation, leading to its superior performance.
> >
> > ---
> > We have incorporated all new experiments in the revised manuscript. Specifically, the comprehensive 32B results can be found in the main Table 1, and the result of generalization to GRPO can be found in Section 5.2 and Table 5. Thank you again for your time and valuable feedback.

---

> > > ### Comment · Reviewer_paED · 2025-11-24
> > > **Response to Rebuttal**
> > >
> > > Thanks for addressing my concerns. I am mostly satisfied with the responses, but could you clarify what k is in W2?

---

> > > > ### Author Response · Authors · 2025-11-25
> > > > **Response to Reviewer paED**
> > > >
> > > > Thank you for the quick follow-up.
> > > >
> > > > $k$ represents the percentage of tokens retained for the policy update. For example, $k=0.4$ means we select the **top 40%** of tokens with the highest scores (entropy or visual dependency) to compute the policy gradient. For the "Random" baseline, $k=0.4$ means 40% of the tokens in the trajectory are randomly selected to participate in the update.
> > > >
> > > > In paper [1], they adopted $k=0.2$, i.e. the top 20% of tokens with the highest entropy scores, to compute the policy gradient. In our table, we implemented [1] across a spectrum of $k$ values from $0.2$ to $0.8$ to ensure we compared against its optimal performance. Therefore, the results clearly show that filtering gradients using Visual Dependency is more effective than using predictive entropy for multimodal reasoning.
> > > >
> > > > We have updated both the Table 4 caption and the corresponding text in Section 5.2 to explicitly define $k$ as the token retention ratio. Thank you again for your time and valuable feedback.
> > > >
> > > > [1] Beyond the 80/20 Rule: High-Entropy Minority Tokens Drive Effective Reinforcement Learning for LLM Reasoning

---

> > > > > ### Comment · Reviewer_paED · 2025-11-25
> > > > > **Response to Rebuttal part 2**
> > > > >
> > > > > Thanks for clarifying it. However, I feel the discussion for Q2 needs to be in the paper, since it seems that makes VPPO different from prior art. Specifically, part 2 of the response about Visually Grounded Tokens.

---

> > > > > > ### Author Response · Authors · 2025-11-26
> > > > > > **Response to Reviewer paED**
> > > > > >
> > > > > > We appreciate this constructive comment, as incorporating the specific discussion on Visually Grounded Tokens effectively underscores the unique advantage of VPPO over prior art.
> > > > > >
> > > > > > Accordingly, we have integrated this analysis into Section 5.2 (under the paragraph "Superiority over Entropy-based Token Selection") of the revised manuscript. The added text clarifies that while entropy-based methods overlook confident (i.e., low-entropy) visual facts, VPPO successfully identifies them as indispensable premises for reasoning.
> > > > > >
> > > > > > We thank you again for this suggestion, which has further improved the clarity and depth of our work.

---

> ### Comment · Reviewer_paED · 2025-11-26
>
> Thanks for the clarification, I have decided to increase my score in response to the rebuttal

---

> > ### Author Response · Authors · 2025-11-27
> > **Response to Reviewer paED**
> >
> > We sincerely appreciate your positive comments. Thank you for taking the time to review our work.

---

### Meta-Review · Area_Chair_k4nJ · 2025-12-27

**Summary:**

This paper addresses a key oversight in multimodal Reinforcement Learning with Verifiable Rewards (RLVR): the neglect of visual perception in optimizing Large Vision-Language Models (LVLMs). Through granular analysis of Chain-of-Thought (CoT) processes, the authors uncover two critical insights: token perception (visual dependency of generated tokens) is sparsely distributed in rollout trajectories, and trajectories vary significantly in overall visual dependency. Building on these observations, they propose Visually-Perceptive Policy Optimization (VPPO), a novel policy gradient algorithm that refines the learning signal via dual mechanisms—reweighting trajectory advantages by visual dependency and focusing updates on perceptually pivotal tokens. VPPO delivers substantial gains over leading open-source RL-tuned models across eight perception and reasoning benchmarks, with consistent validation on 7B and 32B model scales.

Reviewers’ core concerns informing the decision included:
* Sensitivity of the proposed method to the Entropy Loss parameter used in DAPO (Reviewer paED).
* Direct comparison with entropy-based method (Reviewer paED).
* Incremental improvement over PAPO (Reviewer iQRs).
* Additional computational cost (Reviewer 4piz).

Overall, reviewers consistently have positive scores (6,6,6,6) and increase them to (6,6,8,8) since most of the concerns are addressed post-rebuttal. Therefore, I recommend acceptance.

**Reviewer Concerns:**

Most of the concerns are addressed by the author's rebuttal.

**Reviewer Scores:**

Both Reviewer paED and Reviewer iQRs increased their score from 6 to 8 during the discussion. Reviewer CS1Y and Reviewer 4piz might also raise their ratings since their concerns are addressed too.

---

### Decision · Program_Chairs · 2026-01-26

Accept (Poster)